# Antibacterial T6SS effectors with a VRR-Nuc domain are structure-specific nucleases

Julia Takuno Hespanhol[1†], Daniel Enrique Sanchez-Limache[1†], Gianlucca Gonçalves Nicastro[1], Liam Mead[2], Edgar Enrique Llontop[3], Gustavo Chagas-Santos[1], Chuck Shaker Farah[3], Robson Francisco de Souza[1], Rodrigo da Silva Galhardo[1], Andrew L Lovering[2], Ethel Bayer-Santos[1]*

[1]Departamento de Microbiologia, Instituto de Ciências Biomédicas, Universidade de São Paulo, São Paulo, Brazil; [2]Department of Biosciences, University of Birmingham, Birmingham, United Kingdom; [3]Departamento de Bioquímica, Instituto de Química, Universidade de São Paulo, São Paulo, Brazil

**Abstract** The type VI secretion system (T6SS) secretes antibacterial effectors into target competitors. *Salmonella* spp. encode five phylogenetically distinct T6SSs. Here, we characterize the function of the SPI-22 T6SS of *Salmonella bongori* showing that it has antibacterial activity and identify a group of antibacterial T6SS effectors (TseV1–4) containing an N-terminal PAAR-like domain and a C-terminal VRR-Nuc domain encoded next to cognate immunity proteins with a DUF3396 domain (TsiV1–4). TseV2 and TseV3 are toxic when expressed in *Escherichia coli* and bacterial competition assays confirm that TseV2 and TseV3 are secreted by the SPI-22 T6SS. Phylogenetic analysis reveals that TseV1–4 are evolutionarily related to enzymes involved in DNA repair. TseV3 recognizes specific DNA structures and preferentially cleave splayed arms, generating DNA double-strand breaks and inducing the SOS response in target cells. The crystal structure of the TseV3:TsiV3 complex reveals that the immunity protein likely blocks the effector interaction with the DNA substrate. These results expand our knowledge on the function of *Salmonella* pathogenicity islands, the evolution of toxins used in biological conflicts, and the endogenous mechanisms regulating the activity of these toxins.

## Editor's evaluation

This paper will be of interest to microbiologists studying the molecular mechanisms by which bacteria compete with one another, bacterial physiology and toxins in general. Hespanhol et al. report here the characterization of the antibacterial VRR-Nuc family of type VI secretion system effectors that are endonucleases that display antibacterial activity by inducing DNA double-strand breaks.

## Introduction

Bacteria use a series of antagonistic mechanisms to counteract competitors. These processes either require physical contact between attacker and target cells or function in a contact-independent manner via soluble molecules secreted into the medium (*Peterson et al., 2020*). The type VI secretion system (T6SS) is a multiprotein contractile nanomachine evolutionarily related to bacteriophages (*Leiman et al., 2009*). This system is widespread in Gram-negative bacteria and secretes toxic effectors into target cells in a contact-dependent manner (*Coulthurst, 2019*). The T6SS is composed of three major complexes: the membrane complex, the baseplate, and the tail (*Nguyen et al., 2018*). The tail has

*For correspondence:
ebayersantos@usp.br

†These authors contributed equally to this work

Competing interest: The authors declare that no competing interests exist.

a spear-like shape and is propelled against target cells upon a contraction event (*Wang et al., 2017*; *Salih et al., 2018*). The tail tube is composed of hexameric rings of Hcp (hemolysin coregulated protein) capped with a spike composed of a trimer of VgrG (valine–glycine repeat protein G) and a PAAR protein (proline–alanine–alanine–arginine repeats) (*Mougous et al., 2006*; *Shneider et al., 2013*; *Renault et al., 2018*). The effectors secreted via T6SSs associate with Hcp, VgrG, or PAAR either directly or indirectly via adaptor proteins (cargo effectors). In addition, so-called evolved effectors are fused to the C-terminus of Hcp, VgrG, or PAAR (*Cianfanelli et al., 2016*; *Jana and Salomon, 2019*). Several isoforms of VgrG, Hcp, and PAAR proteins can be encoded in the same bacterial genome, usually outside of the T6SS structural gene cluster (and are thus named orphans). These Hcp, VgrG, and PAAR proteins can assemble in different combinations to secrete specific subsets of effectors (*Hachani et al., 2014*; *Bondage et al., 2016*).

T6SSs effectors can target eukaryotic cells, prokaryotic cells or contribute to the acquisition of micronutrients (*Coulthurst, 2019*). The variety of targets is related to the diversity of biochemical activities of T6SS effectors, which can be nucleases, peptidoglycan hydrolases, lipases, NADases, pore-forming proteins, or enzymes that post-translationally modify target proteins (*Jurėnas and Journet, 2021*). Antibacterial effectors with nuclease activity are among the most potent weapons used by an attacker to poison target cells. Several T6SS effectors with nuclease activity have been reported including *Dickeya dadantii* RhsA-CT and RhsB-CT (*Koskiniemi et al., 2013*), *Agrobacterium tumefaciens* Tde1 and Tde2 (*Ma et al., 2014*), *Pseudomonas aeruginosa* PA0099 (*Hachani et al., 2014*), TseT (*Burkinshaw et al., 2018*), and Tse7 (*Pissaridou, 2018*), *Serratia marcescens* Rhs2 (*Alcoforado Diniz and Coulthurst, 2015*), *Escherichia coli* Hcp-ET1, -ET3, and -ET4 (*Ma et al., 2017a*), and Rhs-CT3, -CT4, -CT5, -CT6, -CT7, and -CT8 (*Ma et al., 2017b*), *Acinetobacter baumannii* Rhs2-CT (*Fitzsimons et al., 2018*), *Vibrio parahaemolyticus* PoNe (*Jana et al., 2019*), *Aeromonas dhakensis* TseI (*Pei et al., 2020*), and *Burkholderia gladioli* TseTBg (*Yadav et al., 2021*).

The majority of the nuclease domains mentioned above have been previously predicted by a seminal *in silico* study using comparative genomics (*Zhang et al., 2012*). Among those characterized are Ntox15 (PF15604) (*Ma et al., 2014*), Ntox30 (PF15532), Ntox34 (PF15606), and Ntox44 (PF15607) (*Ma et al., 2017a*), Tox-REase-1 (*Jana et al., 2019*), Tox-REase-3 (PF15647) (*Ma et al., 2017a*), Tox-REase-5 (PF15648) (*Burkinshaw et al., 2018*; *Yadav et al., 2021*), Tox-GHH2 (PF15635) (*Hachani et al., 2014*; *Pissaridou, 2018*), HNH (PF01844) (*Koskiniemi et al., 2013*; *Alcoforado Diniz and Coulthurst, 2015*; *Ma et al., 2017b*), Tox-JAB-2 (*Ma et al., 2017a*), AHH (PF14412) (*Ma et al., 2017a*; *Fitzsimons et al., 2018*), and Tox-HNH-EHHH (PF15657) (*Pei et al., 2020*). However, for most of these studies, the exact nucleotide sequence or structure that is targeted by the effector was not determined.

In *Salmonella* species, T6SSs are encoded in five distinct *Salmonella* pathogenicity islands (SPI-6, SPI-19, SPI-20, SPI-21, and SPI-22) acquired by different horizontal gene transfer events (*Blondel et al., 2009*; *Bao et al., 2019*). The *S. enterica* serovar Typhimurium SPI-6 T6SS is involved in competition with the host microbiota and gut colonization (*Pezoa et al., 2014*; *Brunet et al., 2015*; *Sana et al., 2016*; *Sibinelli-Sousa et al., 2020*) whereas the SPI-19 T6SS of *S.* Gallinarum is involved in survival within macrophages (*Blondel et al., 2013*; *Schroll et al., 2019*). So far, only two T6SS effectors have been characterized in *Salmonella* spp., both targeting peptidoglycan: Tae4 (type VI amidase effector 4) is a gamma-glutamyl-D,L-endopeptidases that cleaves between D-$i$Glu$^2$ and $m$DAP$^3$ within the same peptide stem (*Russell et al., 2012*; *Benz et al., 2013*; *Zhang et al., 2013*) and Tlde1 (type VI L,D-transpeptidase effector 1), which exhibits both L,D-carboxypeptidase and L,D-transpeptidase D-amino acid exchange activity, cleaving between $m$DAP$^3$ and D-Ala$^4$ of the acceptor tetrapeptide stem or replacing the D-Ala$^4$ by a noncanonical D-amino acid, respectively (*Sibinelli-Sousa et al., 2020*).

Herein we report the characterization of the SPI-22 T6SS of *Salmonella bongori* that displays antibacterial activity, and of a group of antibacterial effectors secreted by this system that contain a VRR-Nuc (virus-type replication-repair nuclease) domain (*Kinch et al., 2005*; *Iyer et al., 2006*), which has never been reported to play a role in biological conflicts – named type VI effector VRR-Nuc 1–4 (TseV1–4). TseV1–4 are encoded next to DUF3396-containing proteins, which function as immunity proteins (TsiV1–4) specific to each effector. Phylogenetic analysis revealed that TseVs effectors form a group with other antibacterial effectors belonging to the PD-(D/E)xK phosphodiesterase superfamily. This toxic clade is phylogenetically related to enzymes containing the VRR-Nuc domain involved in DNA repair and metabolism. TseV2 and TseV3 are toxic in *E. coli*, induce DNA double-strand breaks

and activate the SOS response. TseV3 is a $Mn^{2+}$-dependent nuclease that specifically cleaves Y-shaped DNA substrates resembling replication forks or transcription bubbles but not linear ssDNA or dsDNA. Our crystal structure of the TseV3:TsiV3 complex reveals that the immunity protein likely impairs effector toxicity by interacting with and occluding its DNA-binding site. Our results provide mechanistic knowledge about a new group of antibacterial toxins that coopted the VRR-Nuc domain for a previously undescribed role in bacterial antagonism, and further reveal the mode of neutralization via specific immunity protein complexation.

## Results

### The SPI-22 T6SS of *S. bongori* has antibacterial activity

The SPI-22 T6SS of *S. bongori* is phylogenetically related to the HSI-III (Hcp secretion island III) T6SS of *P. aeruginosa* (amino acid similarity ranging from 26% to 80%), and the CTS2 (*Citrobacter rodentium* T6SS cluster 2) of *C. rodentium* (amino acid similarity ranging from 63% to 93%) (*Petty et al., 2010*; *Fookes et al., 2011*; *Figure 1A*). Besides the structural T6SS components encoded within SPI-22, the genome of *S. bongori* NCTC 12419 encodes several orphan proteins comprising two VgrG (SBG_2715, SBG_3770), four Hcp (SBG_0599, SBG_3120, SBG_3143, SBG_3925), three DUF4150/PAAR-like proteins (SBG_1846, SBG_2718, SBG_2955), two adaptors containing DUF2169 (SBG_1847, SBG_2721), and one adaptor with DUF1795 (SBG_3173) (*Figure 1B*).

To analyze whether *S. bongori* SPI-22 T6SS has antibacterial activity, we performed bacterial competition assays using the wild-type (WT) or T6SS null mutant (Δ*tssB/SBG_1238*) strains as attacker cells, and *E. coli* K12 W3110 as prey. Results showed that the prey recovery rate was higher when coincubation was performed with Δ*tssB* compared to the WT (*Figure 1C*). In addition, competition with a Δ*tssB* strain complemented with a plasmid expressing TssB restored the WT phenotype (*Figure 1C*). These results show that the SPI-22 T6SS of *S. bongori* is active in the conditions tested and contributes to interbacterial antagonism, thus priming investigation to further characterize this activity.

### TseV2 and TseV3 are antibacterial SPI-22 T6SS effectors

After verifying that the SPI-22 T6SS has antibacterial activity, we set out to identify the effectors contributing to the antagonistic effect. Initially, we performed *in silico* analysis using Bastion6 (*Wang et al., 2018*) to evaluate several candidates (10 genes up- and downstream of all T6SS components) (*Figure 1B*) for their probability of being a T6SS effector (cutoff score ≥0.5) (data not shown). Two candidates called our attention: SBG_2718 (TseV1) and SBG_2723 (TseV2), which contain an N-terminal PAAR-like domain and a C-terminal VRR-Nuc domain (*Figure 2A*; *Kinch et al., 2005*; *Iyer et al., 2006*). Both putative effectors are encoded next to pairs of genes encoding DUF3396-containing proteins that resemble putative immunity proteins: SBG_2719/TsiV1.1 and SBG_2720/TsiV1.2, and SBG_2724/TsiV2.1 and SBG_2725/TsiV2.2 (*Figure 2A*). Additional BLASTP searches in the genome of *S. bongori* identified two extra VRR-Nuc-containing proteins (SBG_1841/TseV3 and SBG_1828/TseV4), but only one of them encodes an N-terminal PAAR-like domain (SBG_1841). Similarly, SBG_1828 and SBG_1841 are encoded upstream of a DUF3396-containing protein (SBG_1829/TsiV4 and SBG_1842/TsiV3) (*Figure 2A*).

To analyze whether these proteins comprise four effector–immunity pairs, we cloned these genes into compatible vectors under the control of different promoters. To evaluate the toxicity of TseV1–4 upon expression in *E. coli*, the C-terminal regions of TseV1–3 and the full-length TseV4 were cloned into the pBRA vector under the control of the $P_{BAD}$ promoter (inducible by l-arabinose and repressed by d-glucose). The putative immunity proteins were cloned into the pEXT22 vector under the control of the $P_{TAC}$ promoter, which is inducible by IPTG (isopropyl β-D-1-thiogalactopyranoside). *E. coli* strains carrying different combinations of pBRA and pEXT22 were serially diluted and spotted onto LB agar plates containing either 0.2% d-glucose or 0.2% l-arabinose plus 200 μM IPTG (*Figure 2B*). Results showed that TseV2 and TseV3 are toxic in the cytoplasm of *E. coli*, whereas TseV1 and TseV4 do not confer toxicity (*Figure 2B*). Coexpression of TseV2 with either TsiV2.1 or TsiV2.2 revealed that only the first immunity protein neutralizes TseV2 toxicity (*Figure 2B*). Similarly, the toxic effect of TseV3 can be neutralized by coexpression with TsiV3 (*Figure 2B*). Coexpression of TseV2 and TseV3 with all combinations of immunity proteins (TsiV1.1, TsiV1.2, TsiV2.1, TsiV2.2, TsiV3, and TsiV4) revealed that the effectors are neutralized only by the specific cognate immunity protein (*Figure 2—figure*

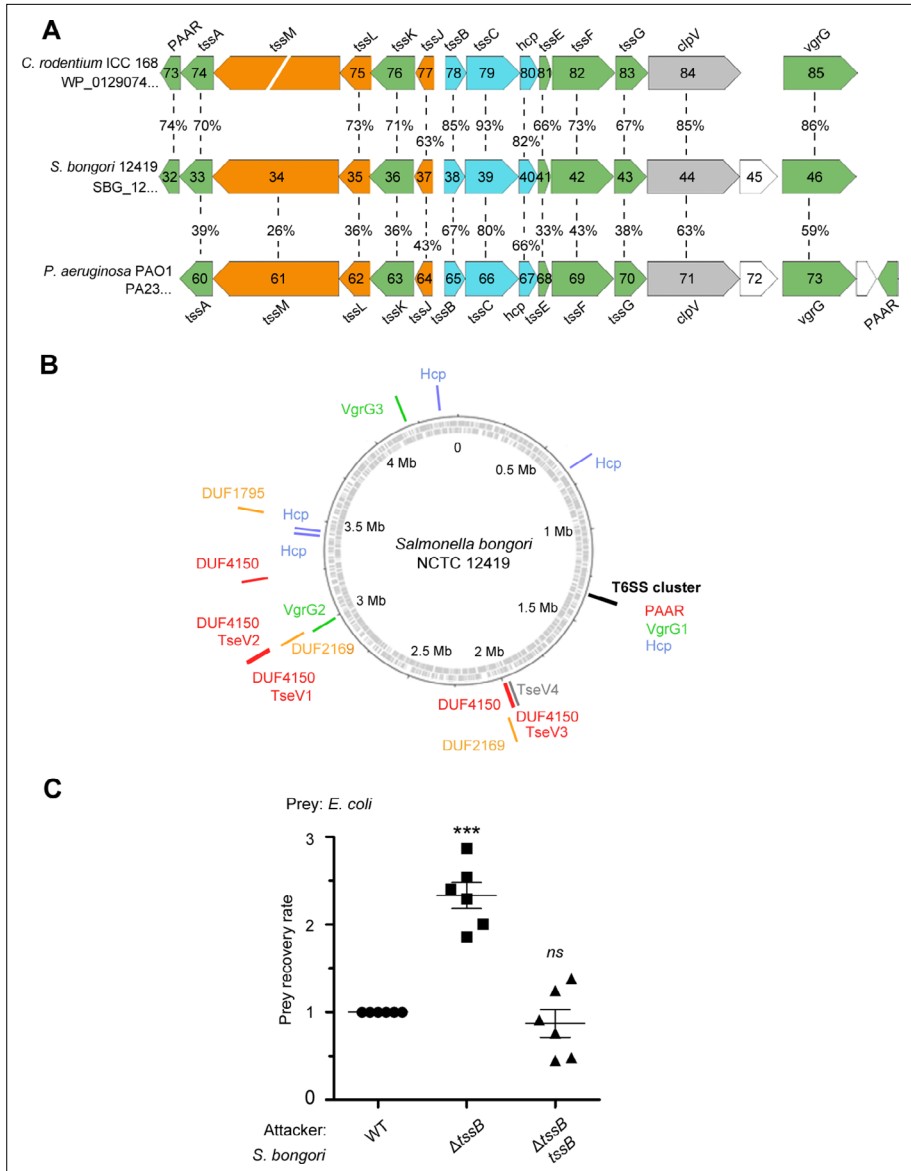

**Figure 1.** The *S. bongori* SPI-22 encodes an antibacterial T6SS. (**A**) Comparison between the SPI-22 T6SS of *S. bongori* with the systems of *C. rodentium* and *P. aeruginosa*. The T6SS proteins forming the three subcomplexes are in colors: membrane components (orange), sheath and inner tube (blue), and baseplate and spike components (green). (**B**) Representation of the circular genome of *S. bongori* with T6SS components highlighted: the structural cluster is marked by a black line; VgrG proteins are represented by green lines; Hcps are in blue; adaptor proteins are in orange; and PAAR or PAAR-like proteins are in red. TseV1, TseV2, and TseV3 fused to PAAR-like domain are also in red, and TseV4 is in gray. (**C**) Bacterial competition assays between *S. bongori* WT, ΔtssB, or ΔtssB complemented with pFPV25.1 *tssB* against *E. coli* in LB agar incubated for 24 hr. The prey recovery rate was calculated by dividing the colony-forming unit (CFU) counts of the output by the input. Data represent the mean ± standard deviation (SD) of six independent experiments and were analyzed through comparison with WT that were normalized to 1. One-way analysis of variance (ANOVA) followed by Dunnett's multiple comparison test. ***p < 0.01 and *ns* (not significant).

The online version of this article includes the following source data for figure 1:

**Source data 1.** CFU counts used to calculate the prey recovery rate of *Figure 1C*.

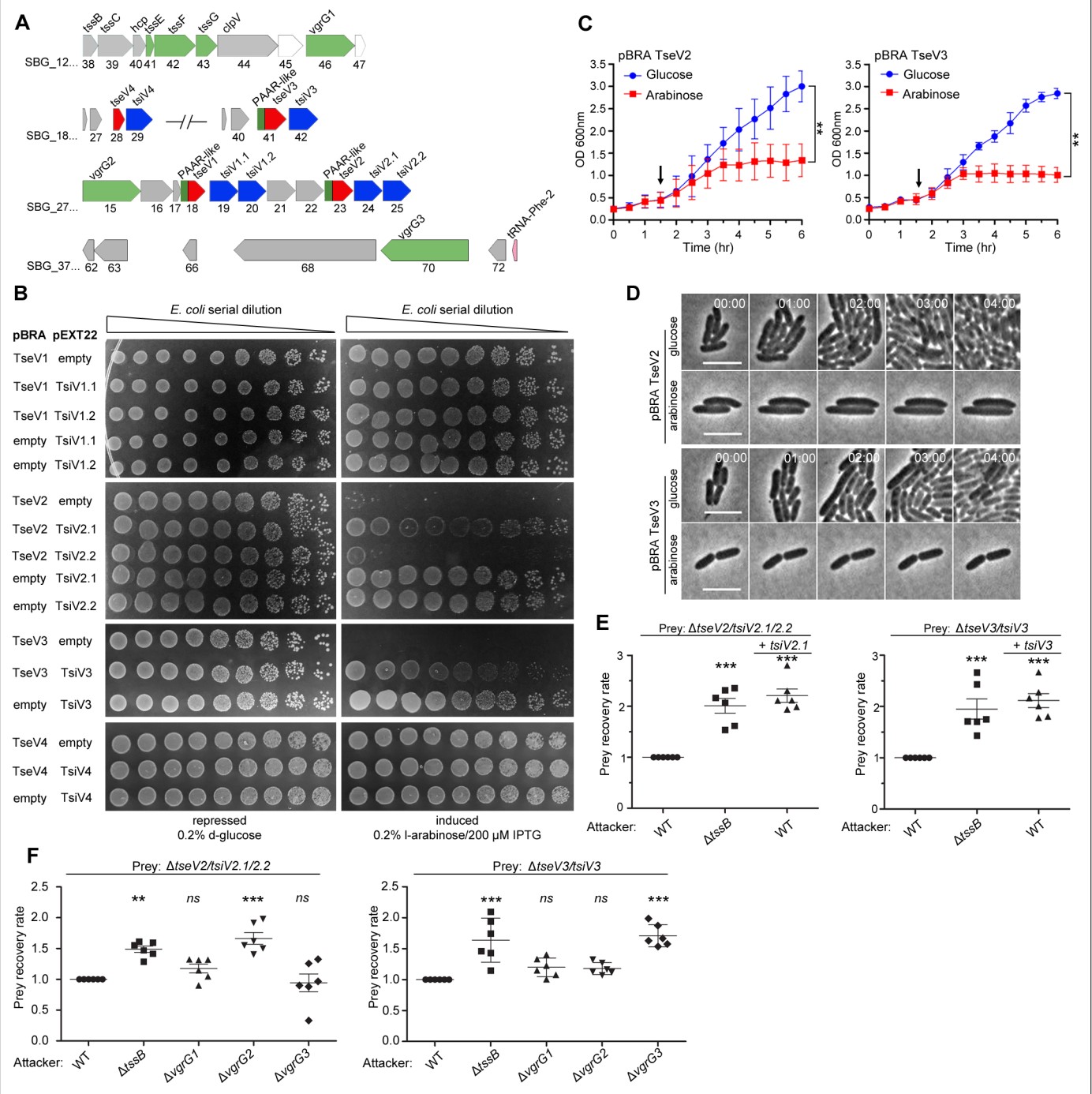

**Figure 2.** TseV2 and TseV3 are antibacterial SPI-22 T6SS effectors. (**A**) Scheme of the genomic region containing VgrGs and TseV/TsiV effector/immunity pairs. VRR-Nuc domain (red), PAAR-like domain (dark green), VgrG (light green), and DUF3396-containing immunities (blue). (**B**) *E. coli* toxicity assay. Serial dilutions of *E. coli* containing pBRA and pEXT22 constructs, as indicated, spotted onto LB agar plates, and grown for 20 hr. Images are representative of three independent experiments. (**C**) Growth curve of *E. coli* harboring pBRA TseV2 or TseV3 before and after toxin induction by the addition of 0.2% l-arabinose (arrow). Results represent the mean ± standard deviation (SD) of three independent experiments performed in duplicate. **p < 0.01 (Student's *t*-test). (**D**) Time-lapse microscopy of *E. coli* carrying either pBRA TseV2 or pBRA TseV3 grown on LB agar pads containing either 0.2% d-glucose (repressed) or 0.2% l-arabinose (induced). Scale bar: 5 µm. Timestamps in hh:mm. (**E**) Bacterial competition assay using *S. bongori* WT, Δ*tssB*, or Δ*tssB* complemented with pFPV25.1 *tssB* against *S. bongori* Δ*tseV2/tsiV2.1/tsiV2.2* or Δ*tseV3/tsiV3* complemented or not with pFPV25.1 *tsiV2.1* or pFPV25.1 *tsiV3*. Strains were coincubated for 20 hr (Δ*tseV2/tsiV2.1/tsiV2.2*) or 6 hr (Δ*tseV3/tsiV3*) prior to measuring CFU counts. The prey recovery rate was calculated by dividing the CFU of the output by the input. Data represent the mean ± SD of six independent experiments and were analyzed through comparison with WT that were normalized to 1. One-way analysis of variance (ANOVA) followed by Dunnett's multiple comparison test. **p <

*Figure 2 continued on next page*

*Figure 2 continued*

0.01, and \*\*\*p < 0.001. (**F**) Bacterial competition assay using *S. bongori* WT, Δ*tssB*, Δ*vgrG1*, Δ*vgrG2*, or Δ*vgrG3* against *S. bongori* Δ*tseV2/tsiV2.1/tsiV2.2* or Δ*tseV3/tsiV3*. Strains were coincubated for 20 hr prior to measuring CFU counts. Prey recovery rate was calculated as in (**E**). Data represent the mean ± SD of six independent experiments. One-way ANOVA followed by Dunnett's multiple comparison test. \*p < 0.05, \*\*p < 0.01, and *ns* (not significant).

The online version of this article includes the following video, source data, and figure supplement(s) for figure 2:

**Source data 1.** Original images of the *E. coli* plates shown in *Figure 2B*.

**Source data 2.** OD$_{600\,nm}$ measures of the growth curve shown in *Figure 2C*.

**Source data 3.** CFU counts used to calculate the prey recovery rate of *Figure 2E, F*.

**Figure supplement 1.** Toxicity assay in *E. coli* cotransformed with pBRA TseV2 or pBRA TseV3 and the six different immunity proteins.

**Figure 2—video 1.** Time-lapse microscopy of *E. coli* harboring pBRA TseV2 growing in media supplemented with 0.2% d-glucose.
https://elifesciences.org/articles/82437/figures#fig2video1

**Figure 2—video 2.** Time-lapse microscopy of *E. coli* harboring pBRA TseV2 growing in media supplemented with 0.2% l-arabinose.
https://elifesciences.org/articles/82437/figures#fig2video2

**Figure 2—video 3.** Time-lapse microscopy of *E. coli* harboring pBRA TseV3 growing in media supplemented with 0.2% d-glucose.
https://elifesciences.org/articles/82437/figures#fig2video3

**Figure 2—video 4.** Time-lapse microscopy of *E. coli* harboring pBRA TseV3 growing in media supplemented with 0.2% l-arabinose.
https://elifesciences.org/articles/82437/figures#fig2video4

*supplement 1*). The effect of TseV2 and TseV3 on cell growth was also analyzed in liquid media by measuring the OD$_{600\,nm}$ of *E. coli* carrying pBRA TseV2 or TseV3 (*Figure 2C*). Under these conditions, bacteria grew normally in media containing d-glucose; but once l-arabinose was added, the culture stopped growing, and the OD$_{600\,nm}$ stabilized (*Figure 2C*).

We performed time-lapse microscopy to evaluate growth and morphology of individual *E. coli* cells harboring pBRA TseV2 or TseV3. Bacteria grew normally when incubated in LB agar pads containing 0.2% d-glucose (repressed) over a time frame of 8 hr (*Figure 2D*, *Figure 2—video 1*, and *Figure 2— video 3*). However, in the presence of 0.2% l-arabinose (induced) bacteria did not grow and remained mostly morphologically unaltered – displaying a modest increase in cell length (*Figure 2D*, *Figure 2— video 2*, and *Figure 2—video 4*).

To verify whether TseV2 and TseV3 are SPI-22 T6SS substrates, we performed bacterial competition assays using *S. bongori* WT or Δ*tssB* (attacker) versus *S. bongori* lacking either TsiV2.1/2.2 (Δ*tseV2/ tsiV2.1/2.2*) or TsiV3 (Δ*tseV3/tsiV3*) as prey (*Figure 2E*). Results demonstrated that the prey recovery rate was higher when prey cells were coincubated with Δ*tssB* compared to WT (*Figure 2E*). Complementation of preys with a plasmid encoding either TsiV2.1 or TsiV3 increased the prey recovery rate, showing that prey became immune to the TseV2- and TseV3-induced toxicity (*Figure 2E*). These results confirm that TseV2 and TseV3 are antibacterial effectors secreted by the SPI-22 T6SS.

As TseV2 and TseV3 contain an N-terminal PAAR-like domain, which interacts with VgrG during T6SS assembly and effector secretion (*Shneider et al., 2013*), we decided to determine which of the three VgrG proteins encoded in the *S. bongori* genome (*Figures 1B and 2A*) were responsible for the secretion of TseV2 and TseV3. To shed light on this matter, we performed bacterial competition assays using *S. bongori* WT, Δ*tssB*, Δ*vgrG1* (SBG_1246), Δ*vgrG2* (SBG_2715), or Δ*vgrG3* (SBG_3770) (attacker) versus Δ*tseV2/tsiV2.1/2.2* or Δ*tseV3/tsiV3* (prey) (*Figure 2F*). The prey recovery rate of Δ*tseV2/tsiV2.1/2.2* increased when this strain was coincubated with Δ*vgrG2*, suggesting that VgrG2 is responsible for secreting TseV2 into target cells (*Figure 2F*). Conversely, the prey recovery rate of Δ*tseV3/tsiV3* increased when this strain was coincubated with Δ*vgrG3*, suggesting that VgrG3 is responsible for secreting TseV3 into target cells (*Figure 2F*). VgrG2 and VgrG3 are 96.9% identical in their N-terminal region (VgrG2$_{1–565}$ and VgrG3$_{1–545}$), but display a distinct C-terminal domain with only 26% identity (VgrG2$_{566–709}$ and VgrG3$_{546–728}$) (*Supplementary file 1*), thus suggesting that this region is responsible for cargo selection (*Liang et al., 2021*). Together, these results show that each effector has its own mechanism of secretion, which is dependent on distinct VgrGs.

# VRR-Nuc-containing effectors are evolutionarily related to Holliday junction resolvases and enzymes involved in DNA interstrand crosslink repair

TseV2 and TseV3 contain a VRR-Nuc domain at their C-terminus, which was initially annotated as DUF994 (*Kinch et al., 2005*) and later renamed VRR-Nuc due to its association with enzymes linked to DNA metabolism (*Iyer et al., 2006*). VRR-Nuc-containing proteins are found in a wide range of organisms, including bacteria, bacteriophages, fungi, and eukaryotes (*Iyer et al., 2006*). Proteins containing this domain comprise a family (PF08774) belonging to the PD-(D/E)xK superfamily, which constitutes a large and functionally diverse group containing representatives involved in DNA replication (Holliday junction resolvases), restriction–modification, repair, and tRNA-intron splicing (*Steczkiewicz et al., 2012*). Members of this superfamily exhibit low sequence similarity but display a common fold in their enzymatic core (with $\alpha_1\beta_1\beta_2\beta_3\alpha_2\beta_4$ topology), which contains conserved residues (Asp, Glu, and Lys) responsible for catalysis (*Steczkiewicz et al., 2012*).

To gain insight into the molecular function of TseV2 and TseV3 and understand their phylogenetic relationship, we used TseV1, TseV2, and TseV3 (TseV4 is 79.1% identical to TseV3 and was not used) amino acid sequences as queries in JackHMMER searches (*Potter et al., 2018*) for four iterations on the NCBI nr database (November 4, 2021) to fetch a total of 2254 sequences with significant similarity (inclusion threshold $\leq 10^{-9}$ and reporting threshold $\leq 10^{-6}$). Additional JackHMMER searches were performed using selected VRR-Nuc-containing proteins as queries (Bce1019, PmgM, T1p21, KIAA1018, HP1472, and Plu1493) (*Iyer et al., 2006*), and recently reported *bona fide* or putative T6SS effectors that also belong to the PD-(D/E)xK superfamily: TseT (*Burkinshaw et al., 2018*), PoNe (*Jana et al., 2019*), IdrD-CT (*Sirias et al., 2020*), TseTBg (*Yadav et al., 2021*), Aave_0499 (RhsB) (*Pei et al., 2022*), and TseV$^{PA}$ (*Wang et al., 2021*). A total of 39,159 sequences were collected. For each JackHMMER dataset, we produced alignments with representatives from clusters formed by sequences displaying 80% coverage and 50–70% identity. These alignments were manually inspected, and divergent/truncated sequences were removed. We observed that the $\beta_2\beta_3\alpha_2\beta_4$ region of the enzymatic core was more conserved so we used this region for a new multiple sequence alignment to build a phylogenetic tree using maximum likelihood (*Figure 3A*).

The resulting tree is composed of nine main clades, with five of these clades comprising PmgM, T1p21, KIAA1018, Bce1019, and HP1472 that reproduce the classification proposed by *Iyer et al., 2006* in which each of these clades constitutes a subfamily of the VRR-Nuc family (*Figure 3A*, gray; *Supplementary file 2*). Bce1019 subfamily contains the endonuclease I from Bacteriophage T7 (PDB 1M0D) (*Hadden et al., 2002*) and the transposon Tn7 encoded nuclease protein TnsA from *E. coli* (PDB 1F1Z) (*Hickman et al., 2000*) (PDB 1T0F) (*Ronning et al., 2004*). The PmgM subfamily contains a nuclease with the same name from phage P1 (*Iyer et al., 2006*). The T1p21 subfamily contains proteins encoded upstream of helicases (*Iyer et al., 2006*). The KIAA1018 group includes the human Fanconi anemia-associated nuclease 1 (FAN1) (PDB 4REA and PDB 4RIA) (*Kratz et al., 2010*; *Liu et al., 2010*; *MacKay et al., 2010*; *Smogorzewska et al., 2010*; *Wang et al., 2014*; *Zhao et al., 2014*) and its bacterial homolog *Pa*FAN1 (PDB 4R89), which are involved in DNA interstrand crosslink repair (*Gwon et al., 2014*; *Wang et al., 2014*; *Zhao et al., 2014*). Curiously, antibacterial T6SS effectors formed four groups (*Figure 3A*, colors) in which TseV2 and TseV3 clustered with Plu1493 (*Iyer et al., 2006*) and TseV$^{PA}$ (*Wang et al., 2021*), whereas homologs of Aave_0499 (*Pei et al., 2022*), IdrD (*Sirias et al., 2020*), and PoNe (*Jana et al., 2019*) formed separated clades (*Figure 3A*, colors; *Supplementary file 2*). These results indicate that TseV proteins are members of the Plu1493 subfamily (*Iyer et al., 2006*). Conversely, homologs of TseT were too divergent to be grouped in the phylogenetic tree and impaired its reproducibility, thus indicating that they probably have a distinct evolutionary origin (*Figure 3—figure supplement 1*; *Supplementary file 2*).

All T6SS effectors (TseVs, Aave_0499, IdrD, and PoNe), except for TseT homologs, formed a clade with a bootstrap value higher than 75% (*Figure 3A*, colors). The genomic context of TseV/Plu1493 homologs is different from the other VRR-Nuc family members (*Supplementary file 3*). While most of VRR-Nuc members (PmgM, T1p21, KIAA1018, Bce1019, and HP1472) are encoded next to genes involved in DNA metabolism, the gene neighborhood of antibacterial T6SS effectors (TseVs, Aave_0499, IdrD, and PoNe) is enriched in proteins encoding components of the T6SS apparatus, adaptors, and immunity proteins (*Figure 3B*; *Supplementary file 3*). In addition, we observed proteins containing domains of other secretion systems involved in biological conflicts, such as CdiB

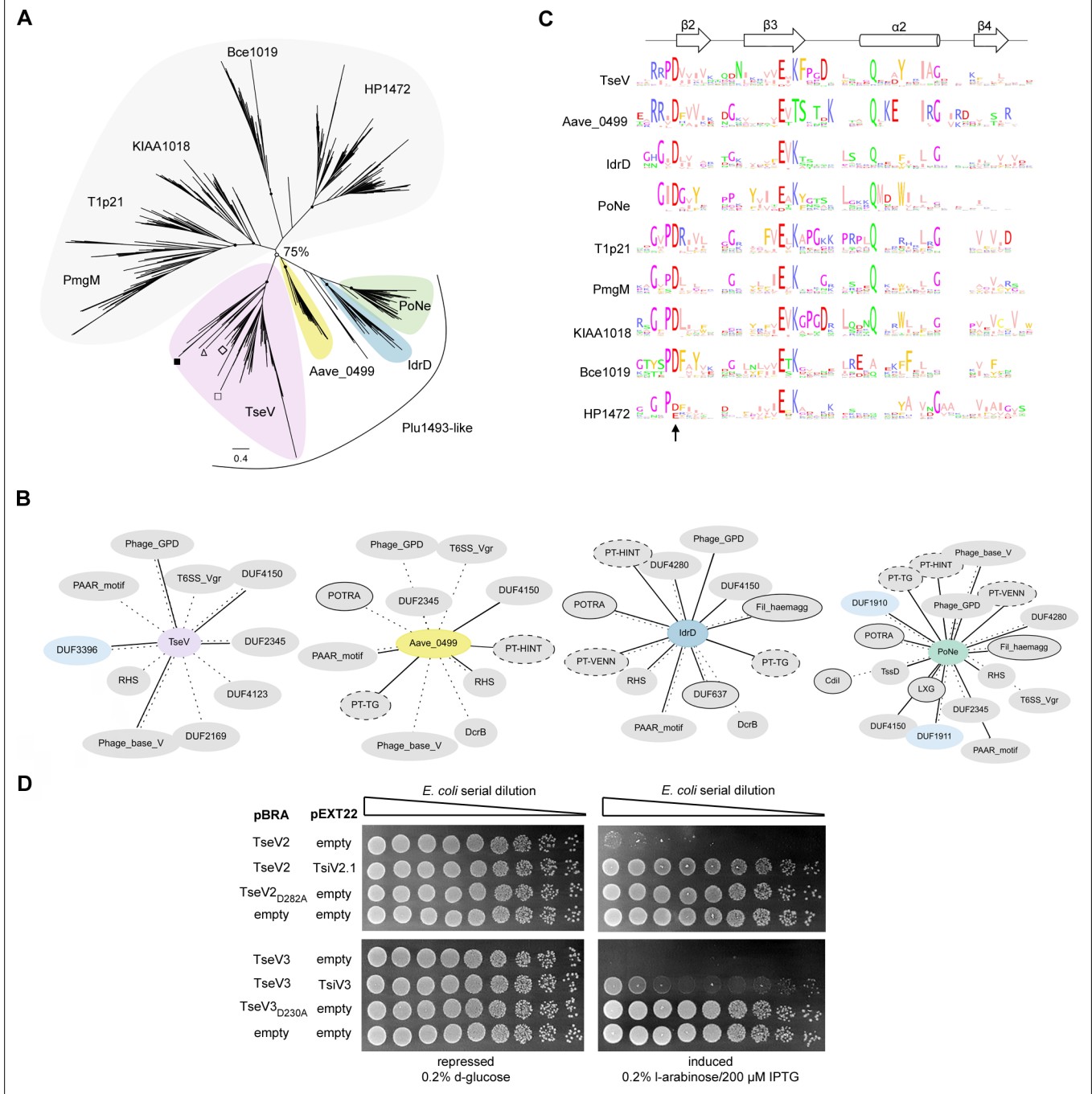

**Figure 3.** VRR-Nuc-containing effectors are evolutionarily related to enzymes involved in DNA repair. (**A**) Maximum-likelihood phylogenetic tree of VRR-Nuc family members (Bce1019, PmgM, T1p21, KIAA1018, HP1472, and Plu1493) (*Iyer et al., 2006*) and recently reported *bona fide* or putative T6SS effectors belonging to the PD-(D/E)xK superfamily (TseT, PoNe, IdrD-CT, TseTBg, Aave_0499, and TseV$^{PA}$). In the TseV clade (pink) the localization of TseV1 (□), TseV2 (△), TseV3 (■), and Plu1493 (◊) are marked. (**B**) Contextual network representation of domains and the genomic context of proteins belonging to Plu1493-like group (TseV, Aave_0499, IdrD, PoNe). Each circle represents a domain, which is either fused to (solid line) or encoded up- or downstream (dashed line) of the gene of interest (center). Borderless gray circles represent domains related to T6SS; bordered gray circles denote domains associated with a different bacterial secretion system; dashed nodes indicate pre-toxin domains; and light blue circles represent immunity proteins. (**C**) Sequence logo from the conserved β$_2$β$_3$α$_2$β$_4$ of the PD-(D/E)xK enzymatic core from all clades shown in (**A**). The arrow indicates conserved aspartic acid that was mutated in (**D**). (**D**) *E. coli* toxicity assay. Serial dilution of *E. coli* containing pBRA and pEXT22 constructs, as indicated, spotted onto LB agar plates and grown for 20 hr. Images are representative of three independent experiments.

The online version of this article includes the following source data and figure supplement(s) for figure 3:

*Figure 3 continued on next page*

*Figure 3 continued*

**Source data 1.** Amino acid sequence alignments used to generate the phylogenetic tree and sequence logos in *Figure 3A, C*.

**Source data 2.** Original images of the *E. coli* plates shown in *Figure 3D*.

**Figure supplement 1.** Comparison of the HMM (Hidden Markov Model) from each clade shown in *Figure 3A*.

and POTRA (T5SS) and LXG (T7SS) (*Figure 3B*; *Supplementary file 3*). Therefore, based on genomic context and biological function, we propose the name Plu1493-like subfamily for the group formed by the clades containing TseVs, Aave_0499, IdrD, and PoNe (*Figure 3A*, colors).

Multiple amino acid sequence alignments from each clade revealed the conserved residues characteristic of the PD-(D/E)xK superfamily (*Figure 3C*), which comprise the aspartic acid (D), glutamic acid (E), and lysine (K) that are part of the catalytic site responsible for hydrolyzing phosphodiester bonds (*Steczkiewicz et al., 2012*). Using this information as a guide, substitution of the conserved aspartic acid for alanine in TseV2 (TseV2$_{D282A}$) and TseV3 (TseV3$_{D230A}$) abrogated toxicity in *E. coli* (*Figure 3D*). These results confirm that the enzymatic activity of the VRR-Nuc domain is essential for toxicity.

## TseV2 and TseV3 induce DNA double-strand breaks and activate the SOS response

We set out to determine whether TseV2 and TseV3 could cause DNA damage by analyzing the activation of the SOS response – a stress response mechanism induced by the activation of RecA (recombinase protein A) in response to DNA damage (*Walker, 1996*). *E. coli* harboring the reporter plasmid pSC101-P$_{recA}$::GFP (*Ronen et al., 2002*), which carries the green fluorescent protein (GFP) under the control of the P$_{recA}$ promoter, was cotransformed with either pBRA TseV2 or TseV3, or their corresponding catalytic mutants as a control (TseV2$_{D282A}$ and TseV3$_{D230A}$), and grown in AB media containing either 0.2% d-glucose or 0.2% l-arabinose (*Figure 4A, B*). We observed an increase in GFP fluorescence when the expression of TseV2 or TseV3 was induced with l-arabinose, indicating the activation of the SOS response (*Figure 4A, B*). GFP expression levels were confirmed by western blot (*Figure 4—figure supplement 1*). To further assess the impact of TseV2 and TseV3 on bacterial chromosome stability, we used DAPI (4′,6-diamidino-2-phenylindole) to stain *E. coli* cells after inducing the expression of TseV2 or TseV3 for 1 hr and evaluated nucleoid integrity by measuring the mean DAPI fluorescence per cell (*Figure 4C–F*). Cells expressing TseV2 or TseV3 revealed smaller/degraded nucleoids and displayed reduced DAPI fluorescence (*Figure 4C–F*).

To evaluate whether TseV2 and TseV3 were degrading *E. coli* DNA, we extracted plasmid DNA after inducing the expression of the WT or catalytic mutant versions. Results revealed a modest degradation in the WT compared with the mutant in induced conditions (*Supplementary file 4*), suggesting that a small number of sites were being cleaved. To increase sensitivity and detect these small number of cleavage sites, we used a reporter double-strand break assay that employs *E. coli* strain SMR14354 encoding a chromosomal GFP fused to the Gam protein from bacteriophage Mu (GamGFP) under the control of the P$_{tet}$ promoter (induced by tetracycline) (*Shee et al., 2013*). The Gam protein binds with high affinity and specificity to DNA double-strand ends, thus inducing the formation of GFP foci at specific sites (*Shee et al., 2013*). *E. coli* SMR14354 carrying an empty pEXT20 plasmid or encoding either TseV2 or TseV3 were grown with 0.2% d-glucose (repressed) or with 200 µM IPTG (induced) and examined by fluorescence microscopy (*Figure 4G, H*). Cells carrying an empty plasmid revealed an even distribution of GamGFP in the cytoplasm, with only a few foci representing spontaneous double-strand breaks (*Figure 4G, H*). Conversely, *E. coli* expressing either TseV2 or TseV3 revealed several intense GFP foci in more than 80% and 75% of cells, respectively (*Figure 4G, H*). The expression of the catalytic mutants TseV2$_{D282A}$ and TseV3$_{D230A}$ did not induce GFP foci formation, indicating that the observed phenotype is specific to the effector's enzymatic activity (*Figure 4G, H*). Interestingly, the expression of TseV2 leads to the formation of fewer intense foci per cell, whereas the expression of TseV3 induces the development of several less intense foci per cell (*Figure 4G, H*), suggesting that TseV3 might cleave DNA at more sites than TseV2. Together, these results suggest that TseV2 and TseV3 cause target cell death by inducing DNA double-strand breaks.

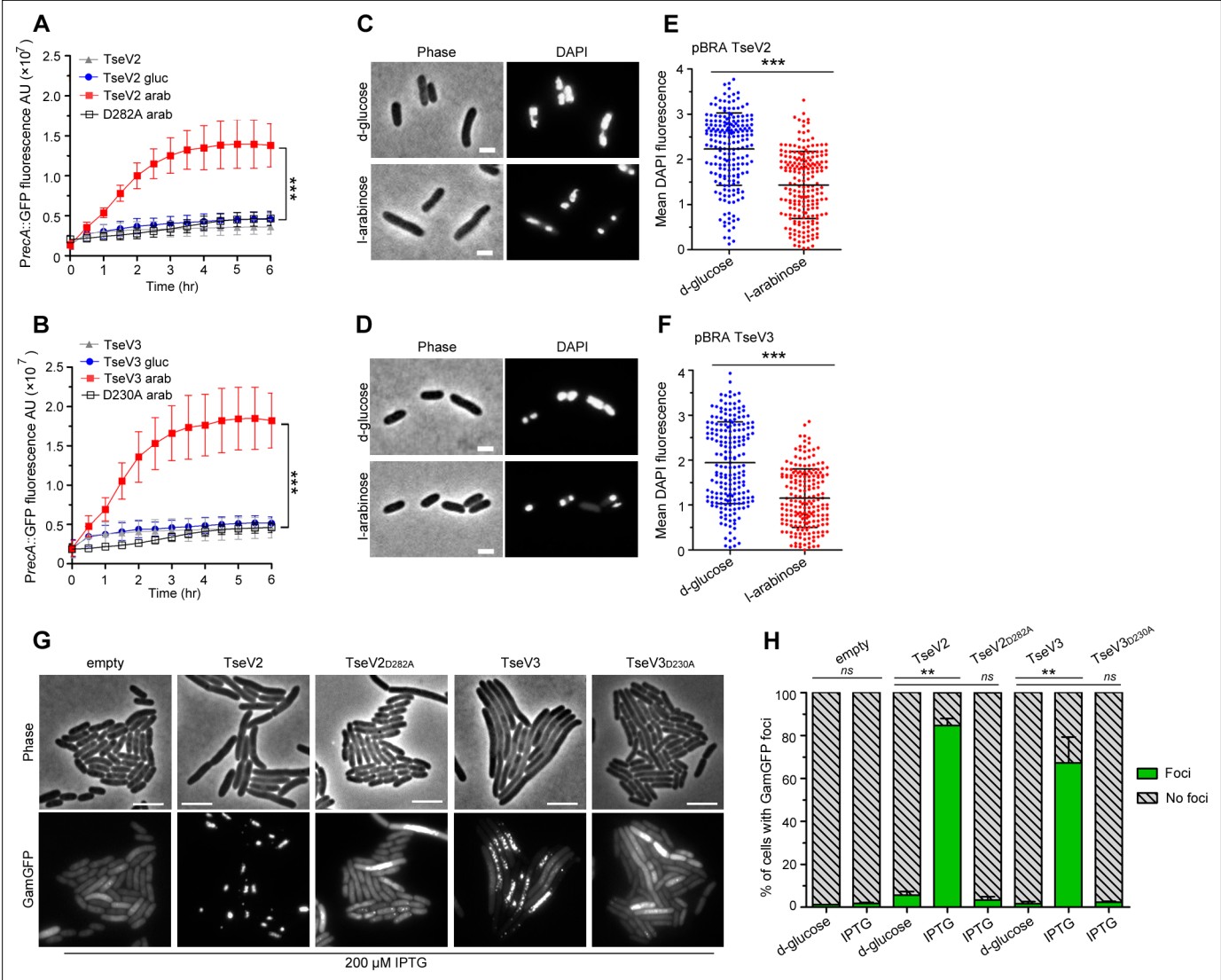

**Figure 4.** TseV2 and TseV3 induce DNA double-strand breaks. Activation of the SOS response was analyzed using *E. coli* cells harboring the reporter plasmid pSC101-P$_{recA}$::GFP and pBRA TseV2 (**A**) or pBRA TseV3 (**B**), which were grown in AB defined media with d-glucose or l-arabinose. Data is the mean ± standard deviation (SD) of three independent experiments. ***p < 0.001 (Student's *t*-test). Bright-field and DAPI images of *E. coli* cells carrying pBRA TseV2 (**C**) or pBRA TseV3 (**D**) grown in the presence of d-glucose (repressed) or l-arabinose (induced). Results are representative images of three independent experiments. (**E, F**) Quantification of the mean 4',6-diamidino-2-phenylindole (DAPI) fluorescence per cell of 200 cells. Data correspond to the mean ± SD of a representative experiment. Scale bar 2 μm. ***p < 0.001 (Student's *t*-test). (**G**) Representative bright-field and GFP images of *E. coli* coexpressing GamGFP and pEXT20 TseV2 or pEXT20 TseV3. Double-strand breaks appear as foci of GamGFP. Images are representatives of three independent experiments. Scale bar: 5 μm. (**H**) Quantification of the GamGFP foci shown in (**G**). Data are shown as the mean ± SD of the three independent experiments. **p < 0.01 (Student's *t*-test).

The online version of this article includes the following source data and figure supplement(s) for figure 4:

**Source data 1.** Values of GFP signal acquired for the SOS response experiment shown in *Figure 4A, B*.

**Source data 2.** Values of 4',6-diamidino-2-phenylindole (DAPI) fluorescence measured for each bacterium, and original images used for quantification shown in *Figure 4C–F*.

**Source data 3.** Original images used to count GamGFP foci shown in *Figure 4G, H* and numbers of foci.

**Source data 4.** Original images used to count GamGFP foci in bacteria carrying pEXT20 TseV2 or TseV2$_{D282A}$ in Figure 4G, H.

**Source data 5.** Original images used to count GamGFP foci in bacteria carrying pEXT20 TseV3 or TseV3$_{D230A}$ in Figure 4G, H.

**Source data 6.** Original images used to count GamGFP foci in bacteria carrying empty pEXT20 in Figure 4G, H.

**Figure supplement 1.** Western blot with anti-GFP antibody of protein extracts from *E. coli* carrying reporter plasmid pSC101 P$_{recA}$::GFP and pBRA with the indicated toxins after induction with 0.2% l-arabinose. anti-DnaK antibody was used as loading control.

*Figure 4 continued on next page*

*Figure 4 continued*

**Figure supplement 1—source data 1.** Original images of western blots.

## TseV3 is a Mn$^{2+}$-dependent structure-specific nuclease

VRR-Nuc-containing enzymes have been shown to be specific for certain DNA structures rather than sequences, both in eukaryotes (*Kratz et al., 2010*; *Liu et al., 2010*; *MacKay et al., 2010*; *Smogorzewska et al., 2010*) and prokaryotes (*Gwon et al., 2014*; *Pennell et al., 2014*). In these cases, the prefered DNA structure is a 5′ flap (*Gwon et al., 2014*) – a Y-shaped DNA form in which one of the arms (5′) is a single strand and the other (3′) is a double strand. In both human FAN1 and *P. aeruginosa PaFAN1*, the 5′ flap is cleaved a couple of nucleotides (1–5 nt) downstream from the arms junction and later the opposite strand is cleaved creating a double-strand end (*Gwon et al., 2014*; *Pennell et al., 2014*). Besides endonuclease activity, FAN1 enzymes also display 5′–3′ exonuclease activity (*Kratz et al., 2010*; *Gwon et al., 2014*).

To confirm whether TseVs were able to cleave DNA *in vitro*, we coexpressed and purified the complex TseV3:TsiV3, and after refolding of TseV3, performed enzymatic assays with an array of different oligonucleotide structures resembling intermediates of replication (splayed arm, 5′ flap, 3′ flap, and three-way junction) (*Figure 5—figure supplement 1*). We used oligonucleotide sequences that were first described in the characterization of human FAN1 (*Kratz et al., 2010*), but in our case oligonucleotides were labeled with the fluorophore FAM (6-carboxyfluorescein) at the 5′ end (oligonucleotide F9). Results revealed that TseV3 preferentially cleaves the splayed arm substrate, with some minor activity on the 3′ flap substrate (*Figure 5A*). The activity was specific as no degradation was detected for the catalytic mutant TseV3$_{D230A}$ (*Figure 5A*).

Human FAN1 and *P. aeruginosa PaFAN1* are metal-dependent nucleases that use Mn$^{2+}$ as a cofactor (*Kratz et al., 2010*; *Gwon et al., 2014*). We analyzed the activity of TseV3 on the splayed arm in the presence of MnCl$_2$ and MgCl$_2$, and some inhibitors of metal-dependent nucleases such as the chelating agent EDTA (ethylenediaminetetraacetic acid) and ZnCl$_2$ (*Kratz et al., 2010*). Results revealed that TseV3 requires Mn$^{2+}$ as a cofactor, and its activity was inhibited by EDTA and partially affected by Zn$^{2+}$ (*Figure 5B*). A time-course experiment revealed the band pattern generated by degradation of the splayed arm substrate containing oligonucleotide F9 with its 5′ labeled with FAM (*Figure 5C*). The same band pattern is observed from 5 to 60 min of incubation, with a decrease in the intensity of the uncleaved substrate (60-mer) and an increase in the intensity of degraded products (*Figure 5C*). In agreement with the reported activity of FAN1 and *PaFAN1*, we observed the appearance of a range of fragments around 35-mer, suggesting that TseV3 cleaves the splayed arm at variable distances after the arms junction (*Figure 5C*), thus resembling what is observed for *PaFAN1* in 5′ flap that cleaves the third to fifth nucleotide downstream from the junction (*Gwon et al., 2014*). The presence of additional smaller fragments could reflect further endo or exonuclease activities of TseV3 (*Figure 5C*). Together, these results confirm that TseV3 displays a unique activity and behaves as a structure-specific nuclease.

## TsiV3 interacts with the putative DNA-binding site of TseV3 to neutralize toxicity

To obtain information about the inhibitory mechanism of TsiV3, we coexpressed it with TseV3 and analyzed the purified complex using size-exclusion chromatography coupled to multiple-angle light scattering (SEC-MALS) (*Figure 6—figure supplement 1*). The MALS calculated average mass for the complex was 66.4 ± 3.3 kDa, which is close to the sum of the theoretical values of their monomers: 26.8 and 37.4 kDa for 6xHis-TseV3 and TsiV3, respectively. Sodium dodecyl sulfate–polyacrylamide gel electrophoresis (SDS–PAGE) analysis of the mixture confirmed the presence of 6xHis-TseV3 and TsiV3 (*Figure 6—figure supplement 1*). These results reveal that TseV3 and TsiV form a 1:1 heterodimeric complex.

We were able to obtain crystals of the TseV3:TsiV3 complex, which belong to space group P2$_1$ and diffracted to a moderate resolution of 4 Å (*Supplementary file 6*). Matthews coefficient analysis indicated that two TseV3:TsiV3 complexes would be the most likely composition in the asymmetric unit. We used AlphaFold (*Jumper et al., 2021*) models of TseV3$_{132–281}$ and TsiV3$_{10–327}$ for molecular replacement using Phaser (*McCoy et al., 2007*), which was able to place two copies of each monomer

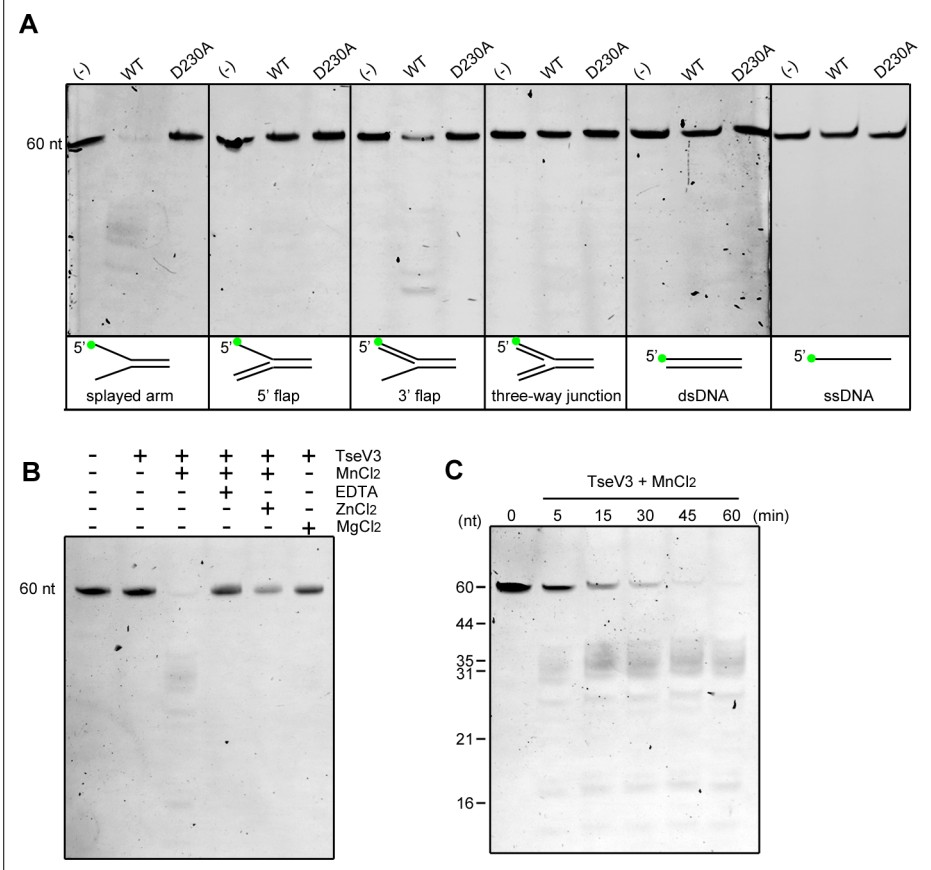

**Figure 5.** TseV3 is a Mn²⁺-dependent structure-specific nuclease. (**A**) *In vitro* enzymatic assay with recombinant TseV3 or TseV3$_{D230A}$ coincubated with different DNA substrates at 37°C for 1 hr. Oligonucleotide F9 was labeled with FAM at the 5' end (green circle). Image is representative of four independent experiments. (**B**) TseV3 was coincubated with splayed arm substrate at 37°C for 1 hr with 5 mM of the indicated cofactors. Image is representative of three independent experiments. (**C**) Time-course degradation of splayed arm by TseV3. Images are representative of three independent experiments.

The online version of this article includes the following source data and figure supplement(s) for figure 5:

**Source data 1.** Original images of enzymatic assays.

**Source data 2.** Original images of cofactors assays.

**Source data 3.** Original images of time-course experiments.

**Figure supplement 1.** Recombinant protein purification and DNA substrates.

**Figure supplement 1—source data 1.** Raw images of sodium dodecyl sulfate–polyacrylamide gel electrophoresis (SDS–PAGE).

**Figure supplement 1—source data 2.** Original images of 20% native and denaturing PAGE.

**Figure supplement 1—source data 3.** Original images of time-course experiment and silver-stained gel.

in the asymmetric unit with a final LLG (log-likelihood gain) of 486.87 and TFZ (translation function Z-score) of 12.4 – with both heterodimeric complexes adopting the same pose (our docked model is available using accession code ma-oyho8 at modelarchive.org). Therefore, the molecular replacement solution using the AlphaFold models most likely represents the correct relative orientation of the two subunits in the TseV3:TsiV3 complex (*Figure 6A*). Given the relatively low resolution of the X-ray diffraction data, we chose not to refine these models against the processed dataset; however, our molecular replacement solution using the AlphaFold models was confirmed by identical placement using experimental PDB homologs taken from the DALI search described below – both TsiV and TsiT can be successfully utilized as search models for our experimental data, producing TFZ scores of 8.3 and 9.2, respectively. Attempts to cofold the TseV3 and TsiV3 complex with AlphaFold did not result

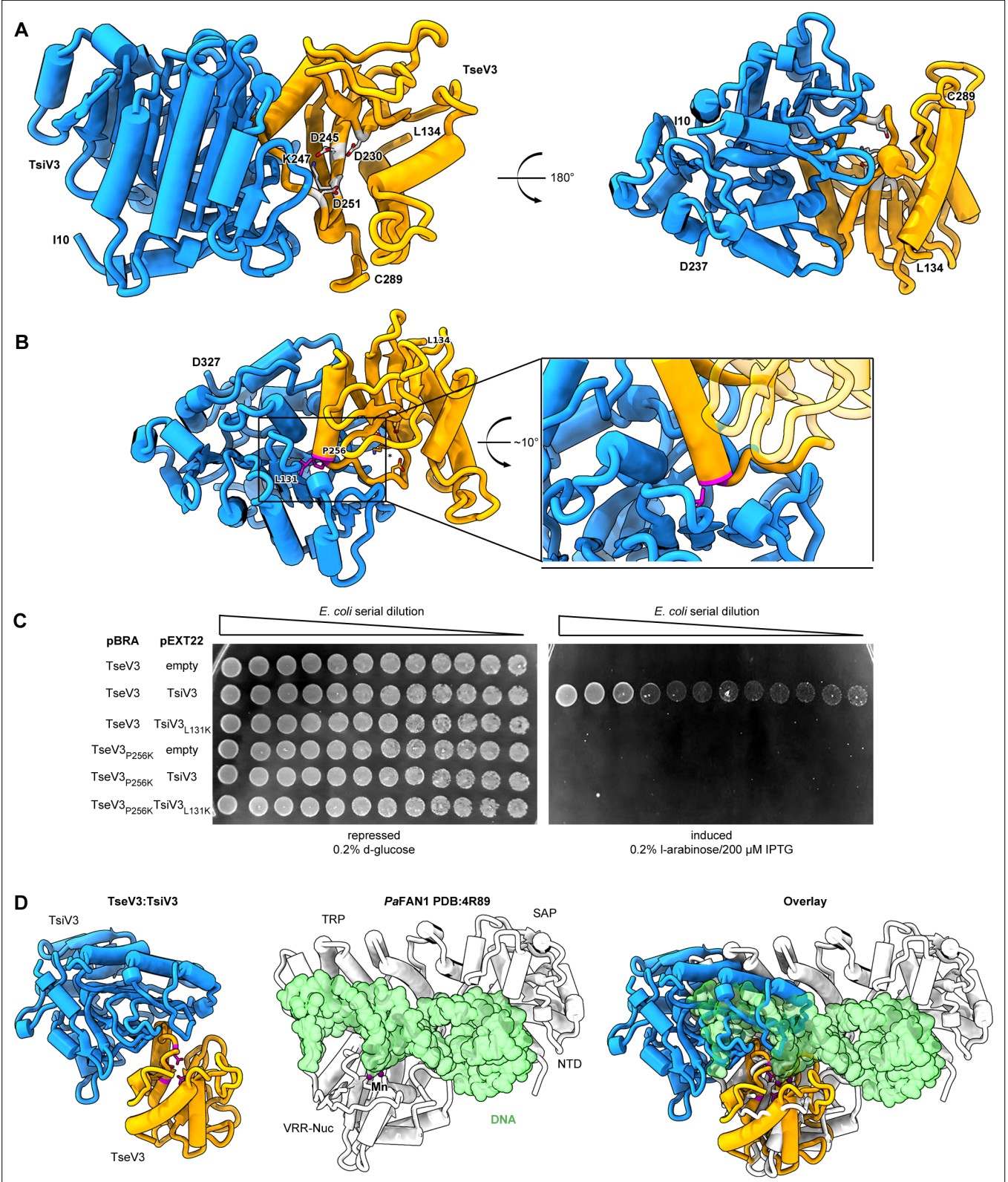

**Figure 6.** The effector–immunity complex reveals that TsiV3 blocks TseV3 substrate-binding site. (**A**) Constrained model of the TseV3:TsiV3 heterodimer with two different views: TsiV3 in blue ($I_{10}$–$D_{327}$) nd TseV3 in orange ($L_{134}$–$C_{289}$). Models are labeled to assist interpretation. PD-(D/E)xK superfamily conserved residues of TseV3 ($D_{230}$, $D_{245}$, and $K_{247}$) are shown in stick form and colored light gray, confirming that they converge to form a putative consensus active site. (**B**) Prediction of interface-compromising mutants in the TseV3:TsiV3 heterodimer. TsiV3 (blue) and TseV3 (orange) with

*Figure 6 continued on next page*

*Figure 6 continued*

putative active sites labeled with asterisk. Residues $L_{131}$ of TsiV3 and $P_{256}$ of TseV3 (both in stick form, magenta) form the closest point of contact in the heterodimer and are at the center of a hydrophobic-rich interface. (**C**) *E. coli* toxicity assay using cells carrying plasmids with wild-type or point mutations in TsiV3 ($L_{131}$K) or TseV3 ($P_{256}$K) as a potential means to destabilize the TseV3:TsiV3 complex interaction. (**D**) Superimposition of the TseV3:TsiV3 coordinates with those of the *Pa*FAN1:DNA complex (PDB 4R89). *Pa*FAN1 protein in white, DNA duplex in green, and catalytic $Mn^{2+}$ are depicted as purple spheres. The overlay (right) is presented in the same orientation as the individual complexes: TseV3:TsiV3 (left, catalytic residues in magenta) and *Pa*FAN1:DNA (middle).

The online version of this article includes the following source data and figure supplement(s) for figure 6:

**Source data 1.** Original image of the *E. coli* plates shown in *Figure 5D*.

**Figure supplement 1.** TseV3:TsiV3 SEC analysis.

**Figure supplement 1—source data 1.** Raw image of the sodium dodecyl sulfate–polyacrylamide gel electrophoresis (SDS–PAGE) with labels and SEC-MALS data.

in the extensive interface we observe in our experimentally docked single models, thus confirming the requirement for data-derived docking.

TsiV3 possesses a central β-sheet that is flanked by α-helices on one side and exposed on the other (*Figure 6A*). This exposed β-sheet surface of TsiV3 binds to an α–β element of TseV3 that flanks its putative active site (composed of residues $D_{230}$, $D_{245}$, and $K_{247}$) (*Figure 6A*). In this configuration, the TseV3-interacting α-helix, which corresponds to the $\alpha_2$ of the classical PD-(D/E)xK $\alpha_1\beta_1\beta_2\beta_3\alpha_2\beta_4$ core topology, projects its N-terminus toward the TsiV3 β-sheet (*Figure 6A*).

To validate the accuracy of our TseV3:TsiV3 structural model, we designed point mutations to interfere with the interaction surface between the two proteins without perturbing the active site or its ability to bind DNA. Thus, residue P256 at the N-terminus of $\alpha_2$ of TseV3 was replaced by lysine (TseV3$_{P256K}$), and residue L131 in the central β-sheet of TsiV3 was also replaced by lysine (TsiV3$_{L131K}$) (*Figure 6B*). Native and mutated versions of the effector and immunity protein were cotransformed in *E. coli* to analyze toxicity (*Figure 6C*). Results revealed that TsiV3$_{L131K}$ was unable to neutralize TseV3 toxicity. In addition, the mutated TseV3$_{P256K}$ maintained its enzymatic activity, displaying toxicity in *E. coli*; however, this mutant was not neutralized by coexpression with the native immunity protein (TsiV3) (*Figure 6C*). Although point mutations did not completely disrupt effector:immunity complex formation (*Figure 6—figure supplement 1*), the interference with the interaction was sufficient to prevent neutralization of the effector in toxicity assays (*Figure 6C*). Together, these results reinforce the accuracy of our model, which is comprised of both experimental constraints and theoretical model components.

Searches for structures similar to TseV3$_{132-281}$ and TsiV3$_{10-327}$ using the DALI server (*Holm, 2020*) revealed matches to proteins of related function (a top *Z*-score of 33.5 for the immunity protein PA0821 PDB:7DRG and TsiV3; and a top *Z*-score of 3.8 between psNUC PDB:4QBL and TseV3). The modest RMSD (root-mean-square deviation) for the Cα positions in TseV3 and other VRR-Nuc enzymes indicates that TseV3 represents a variant of the VRR-Nuc fold. Nevertheless, the PD-(D/E)xK consensus catalytic residues are identifiable as the modified sequence MD$_{230}$IX$_n$D$_{245}$VK$_{247}$ in TseV3. These residues are found in positions commensurate with active nucleases of the PD-(D/E)xK superfamily. Accordingly, superimposition of TseV3 with the well-characterized VRR-Nuc member *Pa*FAN1 (PDB 4R89) (*Gwon et al., 2014*) (clade KIAA1018 in *Figure 3A*) matches residues $D_{230}$, $D_{245}$, and $K_{247}$ of the former with residues $D_{507}$, $E_{522}$, and $K_{524}$ of the latter (*Figure 6D*).

*Pa*FAN1 is the bacterial homolog of human FAN1 (*MacKay et al., 2010*), which is involved in the repair of DNA interstrand crosslinks. Human FAN1 is comprised of four domains: ubiquitin-binding zinc (UBZ); SAF-A/B, Acinus, and PIAS (SAP); tetratricopeptide repeat (TPR); and VRR-Nuc. Conversely, *Pa*FAN1 lacks the UBZ domain and contains an uncharacterized N-terminal domain (NTD), followed by SAP, TPR, and VRR-Nuc (*Iyer et al., 2006*; *Gwon et al., 2014*). The structure of *Pa*FAN1 has been solved in complex with a 5′ flap (PDB 4R89) (*Figure 6D*, middle). As the catalytic residues of TseV3 align with those of *Pa*FAN1, we used the structure of the latter as a guide to analyze the likely mechanism by which TsiV3 may neutralize TseV3 activity. Comparison between the *Pa*FAN1:DNA and TseV3:TsiV3 complexes reveals that the path of the DNA substrate is potentially incompatible with the presence of TsiV3 (*Figure 6D*). Hence, assuming the mode of substrate recognition is similar between *Pa*FAN1 and TseV3, this result suggests that the binding of TsiV3 sterically blocks the toxin active site.

In our model of the complex, TsiV3 occlusion of the TseV3 active site would be enabled by a TsiV3 loop, which projects into the putative TseV3 DNA-binding pocket (*Figure 6A, D*).

## Discussion

Bacterial antagonistic strategies targeting nucleic acids are very effective as these components are critical for life. In this study, we characterized a group of effectors containing the VRR-Nuc domain, which comprise the first example of effectors with a structure-specific nuclease activity. This domain has not previously been reported to be used in biological conflicts (*Zhang et al., 2012*), but recently was suggested to work as a T6SS effector due to its localization next to a PAAR protein in *P. aeruginosa* (*Wang et al., 2021*) – for consistency we decided to keep the name TseV for this group of effectors. Proteins containing the VRR-Nuc domain comprise a family (*Iyer et al., 2006*) belonging to the PD-(D/E)xK superfamily, which contain a conserved enzymatic core composed by $\alpha_1\beta_1\beta_2\beta_3\alpha_2\beta_4$ (*Steczkiewicz et al., 2012*). The conserved catalytic residues (D, E, and K) are located in the central $\beta_2\beta_3$-sheet, while the $\alpha_1$-helix is associated with the formation of the active site and $\alpha_2$-helix with substrate binding (*Steczkiewicz et al., 2012*). Curiously, *S. bongori* encodes four TseV homologs: TseV2 and TseV3 are toxic in *E. coli*, whereas TseV1 and TseV4 are not toxic (*Figure 2*). Based on what is known about the catalytic mechanism of PD-(D/E)xK nucleases, we hypothesize that the lack of $\alpha_2$- and $\alpha_1$-helix in TseV1 and TseV4, respectively, might explain the lack of toxicity (*Supplementary file 5*). Another curiosity is the presence of two homologs of the DUF3396 immunity genes downstream of both TseV2 and TseV1 (*Figure 2A*). Such genomic organization is also conserved in other bacterial species like *Photorhabdus thracensis* (VY86_01065, VY86_01040), *Photorhabdus asymbiotica* (PAU_03539, PAU_03660) (Enterobacterales), *Marinobacter nauticus* (MARHY2492) (Pseudomonadales), and *Herbaspirillum huttiense* (E2K99_00955) (Burkholderiales). The fact that only one immunity protein (TsiV2.1) can neutralize the effector (TseV2) makes us wonder about the role of the additional immunity protein gene – and why such genomic context is conserved in other bacterial species (*Supplementary file 3*). One possibility is that the extra immunity protein could regulate the effector at the transcriptional level as has been reported for the immunity protein TsiTBg known to regulate a different PD-(D/E)xK effector (TseTBg) (*Yadav et al., 2021*). It is worth mentioning that such genomic organization for immunity proteins have been reported for the SUKH superfamily of immunity proteins, which are encoded next to various nucleases (*Zhang et al., 2011*).

The complexity of the PD-(D/E)xK superfamily and the rapid evolution of polymorphic toxins make it difficult to categorize antibacterial effectors belonging to this group. However, our phylogenetic analysis was able to confidently group VRR-Nuc-containing effectors into one clade (TseV) and show that this group is different from the clades formed by the homologs of additional T6SS effectors (Aave_0499, IdrD, and PoNe) (*Figure 3*). Although proteins belonging to clades Aave_0499, IdrD, and PoNe are not recognized by the Pfam model of VRR-Nuc, these proteins share similar genetic architectures concerning domain fusions and gene vicinity (*Figure 3B*); therefore, we decided to call this larger group Plu1493-like subfamily to respect the original nomenclature proposed by *Iyer et al., 2006*.

The enzymatic activity of proteins belonging to the PD-(D/E)xK superfamily is quite diverse, but we were able to narrow down the possibilities and reveal that TseV3 is a structure-specific nuclease that retains the peculiar activity displayed by their related homologs containing the VRR-Nuc domain that work in DNA interstrand crosslink repair (FAN1 and *Pa*FAN1); however, TseV3 diverged in terms of substrate specificity and preferentially cleaves splayed arms instead of 5′ flaps. Given the biochemical evidence suggesting that TseV3 cleaves splayed arms, we hypothesize that TseV3 acts on replication forks or transcription bubbles of target cells. The effector activity could be directly responsible for creating the double-strand breaks detected in the GamGFP reporter assay, which can detect up to a four-base single-strand DNA overhang (*Akroyd and Symonds, 1986*; *Shee et al., 2013*); or its activity on replication forks could promote the collapse of the replication machinery, thus inducing double-strand breaks (*Kuzminov, 1999*).

T6SSs effector–immunity complexes are related to type II toxin–antitoxin (TA) systems, which play several roles in bacterial physiology ranging from genomic stabilization and abortive phage infection to stress modulation and antibiotic persistence (*Fraikin et al., 2020*). Most T6SS immunity proteins described to date bind to effectors to regulate their enzymatic activity (*Benz et al., 2012*; *Benz et al., 2013*; *Dong et al., 2013*; *Li et al., 2013*; *Lu et al., 2014*; *Robb et al., 2016*). An exception is Tri1

(type VI secretion ADP-ribosyltransferase immunity 1) from *Serratia proteamaculans*, which exhibits two modes of inhibition: active site occlusion and enzymatic removal of a post-translational modification (*Ting et al., 2018*). The neutralization mechanism of TsiT, which counteracts the PD-(D/E)xK effector TseT from *P. aeruginosa*, was also proposed to be different: TsiT interferes with the effector oligomerization state and hinders its nuclease activity (*Wen et al., 2021*). Our structural model of the TseV3:TsiV3 complex revealed that TsiV3 β-sheet binds to the α$_2$-helix of TseV3, which is involved in DNA binding in other PD-(D/E)xK members (*Steczkiewicz et al., 2012*). In addition, the superposition of the TseV3:TsiV3 complex with the structure of *Pa*FAN1 bound to DNA reinforces the hypothesis that TsiV3 likely occludes the substrate-binding site of TseV3.

This work adds to the diversity of antibacterial weapons, placing the structure-specific nucleases from the VRR-Nuc family within the remit of antibacterial effectors. Knowledge about the phylogeny and mechanism of action of this group of effectors will be important in interpreting its function in other bacterial species, including the requirements of neutralization by very specific immunity pairings.

## Materials and methods

### Bacterial strains and growth conditions

A list of bacterial strains used in this work can be found in *Supplementary file 7*. Strains were grown at 37°C in Lysogeny Broth (10 g/l tryptone, 10 g/l NaCl, 5 g/l yeast extract) under agitation. Cultures were supplemented with antibiotics in the following concentration when necessary: 50 µg/ml kanamycin, 100 µg/ml ampicillin, and 50 µg/ml streptomycin.

### Cloning and mutagenesis

Putative effectors SBG_1828, SBG_1841, SBG_2718, and SBG_2723 were amplified by PCR and cloned into pBRA vector under the control of P$_{BAD}$ promoter (*Souza et al., 2015*). Immunity proteins SBG_1828, SBG_1842, SBG_2719, SBG_2720, SBG_2724, and SBG_2725 were cloned into pEXT22 under the control of P$_{TAC}$ promoter (*Dykxhoorn et al., 1996*). TseV2 and TseV3 were cloned in the pEX20 vector under the control of P$_{TAC}$ promoter (*Dykxhoorn et al., 1996*) for GamGFP assays. For complementation, SBG_1238 (TssB), SBG_1842 (TsiV3), and SBG_2724 (TsiV2.1) were cloned into pFPV25.1 by replacing the GFP mut3 coding region for the genes of interest (*Valdivia and Falkow, 1996*). Point mutations were created using QuikChange II XL Site-Directed Mutagenesis Kit (Agilent Technologies) and pBRA TseV2 and pBRA TseV3 plasmids as templates. *S. bongori* mutant strains were constructed by $\lambda$-Red recombination engineering using a one-step inactivation procedure (*Datsenko and Wanner, 2000*). All constructs were confirmed by sequencing.

### Interbacterial competition assay

Bacterial competition assays were performed using *S. bongori* (WT, Δ*tssB*, Δ*tseV2/tsiV2.1/tsiV2.2*, or Δ*tseV3/tsiV3*) as attackers, and *E. coli* K-12 W3110 carrying pEXT22 Km$^R$ as prey. Overnight cultures of the attacker and prey cells were subcultured in LB (1:30) until reaching OD$_{600\ nm}$ 1.6, then adjusted to OD$_{600\ nm}$ 0.4 and mixed in a 10:1 ratio (attacker:prey), 5 µl of the mixture were spotted onto 0.22-µm nitrocellulose membranes (1 × 1 cm) and incubated on LB agar (1.5%) at 37°C for the indicated periods. Membranes containing the bacterial mixture were placed on 1.5 ml tubes containing 1 ml of LB, homogenized by vortex, serially diluted, and plated on selective plates containing antibiotics. The prey recovery rate was calculated by dividing the CFUs (colony-forming units) counts of the output by the CFU of the input.

### *E. coli* toxicity assays

Overnight cultures of *E. coli* DH5α (LB with 0.2% d-glucose) carrying effectors (in pBRA) and immunity proteins (in pEXT22) were adjusted to OD$_{600\ nm}$ 1, serially diluted in LB (1:4) and 5 µl were spotted onto LB agar plates containing either 0.2% d-glucose or 0.2% l-arabinose plus 200 µM IPTG – both supplemented with streptomycin and kanamycin – and incubated at 37°C for 20 hr. For growth curves, overnight cultures of *E. coli* carrying pBRA TseV2 or TseV3 were inoculated in LB (1:50) with 0.2% d-glucose and grown at 37°C (180 rpm) for 1.5 hr. Next, media was replaced with either fresh warm LB containing 0.2% d-glucose or 0.2% l-arabinose.

## Time-lapse microscopy

For time-lapse microscopy, LB agar (1.5%) pads were prepared by cutting a rectangular piece out of a double-sided adhesive tape, which was taped onto a microscopy slide as described previously (*Bayer-Santos et al., 2019*). *E. coli* DH5α harboring pBRA TseV2 or TseV3 were subcultured in LB (1:50) with 0.2% d-glucose until reaching $OD_{600\ nm}$ 0.4–0.6 and adjusted to $OD_{600\ nm}$ 1.0. Cultures were spotted onto LB agar pads supplemented either with 0.2% d-glucose or 0.2% l-arabinose plus antibiotics. Images were acquired every 15 min for 16 hr using a Leica DMi-8 epifluorescent microscope fitted with a DFC365 FX camera (Leica) and Plan-Apochromat ×63 oil objective (HC PL APO ×63/1.4 Oil ph3 objective Leica). Images were analyzed using FIJI software (*Schindelin et al., 2012*).

## Bioinformatic analysis

Iterative profile searches using JackHMMER (*Eddy, 2011*) with a cutoff *e*-value of $10^{-6}$ and a maximum of four iterations were performed to search a non-redundant (nr) protein database from the National Center for Biotechnology Information (NCBI) (*Sayers et al., 2019*). Similarity-based clustering of proteins was carried out using MMseqs software (*Steinegger and Söding, 2017*). Sequence alignments were produced with MAFFT (RRID:SCR_011811) local-pair algorithm (*Katoh and Standley, 2013*), and noninformative columns were removed with trimAl software (RRID:SCR_017334) (*Capella-Gutiérrez et al., 2009*). Approximately maximum-likelihood phylogenetic trees were built using FastTree 2 (RRID:SCR_015501) (*Price et al., 2010*). Sequence logos were generated using Jalview (RRID:SCR_006459) (*Waterhouse et al., 2009*). HMM models were produced for each sequence alignment and compared against each other with the HH-suite package (RRID:SCR_016133) (*Steinegger et al., 2019*). Proteins were annotated using the HHMER package (*Eddy, 2011*) or HHPRED software (RRID:SCR_010276) (*Söding et al., 2005*) and Pfam (RRID:SCR_004726) (*Bateman et al., 2004*), PDB (*Berman et al., 2007*), or Scope (*Fox et al., 2014*) databases. An in-house Python script was used to collect the gene neighborhoods based on information downloaded from the complete genomes and nucleotide sections of the NCBI database (*Sayers et al., 2019*).

TseV1–4 sequence alignments were produced with MAFFT local-pair algorithm (*Katoh and Standley, 2014*) and analyzed in AilView (*Larsson, 2014*) to separate the regions of interest. Sequence logos were produced using the Jalview (*Waterhouse et al., 2009*). Protein structure predictions were performed with ColabFold (*Mirdita et al., 2021*) and AlphaFold (*Jumper et al., 2021*), and visualization was performed using Pymol (*DeLano, 2002*).

## SOS response assays

Overnight cultures of *E. coli* MG1655 harboring the reporter plasmid pSC101-$P_{recA}$::GFP (*Ronen et al., 2002*) and pBRA TseV2 and TseV2$_{D282A}$ or TseV3 and TseV3$_{D230A}$ were subcultured (1:50) in LB with 0.2% d-glucose and grown at 37°C until $OD_{600\ nm}$ 0.4–0.6. Bacteria were harvested and resuspended in AB defined media (0.2% $(NH_4)_2SO_4$, 0.6% $Na_2HPO_4$, 0.3% $KH_2PO_4$, 0.3% NaCl, 0.1 mM $CaCl_2$, 1 mM $MgCl_2$, 3 µM $FeCl_3$) supplemented with 0.2% sucrose, 0.2% casamino acids, 10 µg/ml thiamine, and 25 µg/ml uracil (*Bayer-Santos et al., 2019*). Cells ($OD_{600\ nm}$ 1.0) were placed in a black 96-well plate with clear bottom (Costar) with 0.2% d-glucose or 0.2% l-arabinose to a final volume of 200 µl. GFP fluorescence was monitored in a plate reader SpectraMax Paradigm Molecular Devices for 6 hr at 30°C.

## DAPI staining

*E. coli* DH5α carrying pBRA TseV2 and TseV3 were subcultured in LB with 0.2% d-glucose (1:50) and grown at 37 °C (180 rpm) until $OD_{600\ nm}$ 0.4–0.6. Cells were harvested and resuspended in new media with 0.2% d-glucose or 0.2% l-arabinose and growth for an additional 1 hr. Bacteria were fixed with 4% paraformaldehyde for 15 min on ice, washed in phosphate-buffered saline (PBS) and stained with DAPI (3 µg/ml) for 15 min at room temperature. Samples were washed once with PBS before transferring 1 µl of each culture to a 1.5% PBS-agarose pad for visualization. Images were acquired in Leica DMi-8 epifluorescent microscope fitted with a DFC365 FX camera (Leica) and Plan-Apochromat ×63 and ×100 oil objectives (HC PL APO ×63 and ×100/1.4 Oil ph3 objectives Leica). Images were analyzed using FIJI software (*Schindelin et al., 2012*). To assess DNA integrity, the mean pixel fluorescence per cell was manually measured from 200 bacteria from different fields from each experiment. The cell area was determined using the bright field, and the mean pixel fluorescence per cell was measured in the DAPI channel subtracting the background.

## DNA double-strand break assay

*E. coli* SMR14354 containing a chromosomal GamGFP under the control of P$_{tet}$ promotor (*Shee et al., 2013*) and harboring an empty pEXT20 or encoding TseV2 or TseV3 or catalytic mutants TseV2$_{D282A}$ or TseV3$_{D230A}$ were subcultured in LB (1:100) with 0.2% d-glucose grown for 1.5 hr at 37°C (180 rpm) before the induction of GamGFP with 50 ng/ml tetracycline for 2 hr. Bacteria were resuspended in new media with either 0.2% d-glucose or 200 µM IPTG and grown for 1 hr. One microliter of each culture was spotted onto a 1.5% AB agarose pad. Images were acquired in a Leica DMi-8 epifluorescent microscope fitted with a DFC365 FX camera (Leica) and Plan-Apochromat ×100 oil objective (HC PL APO ×100/1.4 Oil ph3 objective Leica). Images were analyzed using FIJI software (*Schindelin et al., 2012*). At least 400 bacteria from each experiment were quantified.

## Protein expression and purification

*E. coli* SHuffle cells carrying pRSFDuet 6xHis-TseV3:TsiV3 were grown in LB supplemented with kanamycin (30°C, 180 rpm) until OD$_{600\ nm}$ 0.4–0.6. Expression was induced with 0.5 mM IPTG followed by incubation at 16°C for 16 hr. Cells were harvested via centrifugation at 9000 × *g* for 15 min, and pellets were resuspended in buffer A (50 mM Tris–HCl pH 7.5, 200 mM NaCl, 5 mM imidazole) and lysed at 4°C using an Avestin EmulsiFlex-C3 homogenizer. The lysate was collected and centrifuged (48,000 × *g*) for 1 hr at 4°C. The supernatant was loaded onto a 5-ml HisTrap HP cobalt column (GE Healthcare) equilibrated in buffer A. The column was washed with 10 column volumes (CV) of buffer A before running an elution gradient of 0–50% buffer B (50 mM Tris–HCl pH 7.5, 200 mM NaCl, 500 mM imidazole) over 10 CV, followed by a final 10 CV wash with 100% buffer B. The presence of TseV3:TsiV3 was confirmed by SDS–PAGE of eluted fractions. TseV3:TsiV3 was concentrated using a Vivaspin spin-concentrator and further purified by size-exclusion chromatography on a Superdex 200 26/60 column (GE Healthcare) equilibrated in 50 mM Tris–HCl pH 7.5, 150 mM NaCl.

SEC-MALS analyses were used to determine the molar mass of the TseV3–TsiV3 complex (concentration 3.2 mg/ml). Protein samples (400 µl injection volume) were separated using a Superdex 200 10/300 column (GE Healthcare) equilibrated with buffer (50 mM Tris–HCl pH 7.5, 20 mM NaCl) coupled to a miniDAWN TREOS multiangle light scattering system and an Optilab rEX refractive index detector. Data analysis was performed using the Astra Software package version 7.1 (Wyatt TechnologyCorp). Molecular mass was calculated assuming a refractive index increment dn/dc = 0.185 ml/g (*Wen et al., 1996*). Fractions were analyzed in SDS–PAGE to confirm protein molecular weight.

For enzymatic assays, *E. coli* BL21(DE3) carrying pRSFDuet 6xHis-TseV3:TsiV3 or 6xHis-TseV3$_{D230A}$:TsiV3 were grown in LB supplemented with kanamycin (37°C, 180 rpm) until OD$_{600\ nm}$ 0.6. Protein expression was induced with 200 µM IPTG at 18°C for 16 hr. Cells were harvested at 9000 × *g* for 15 min, pellets were resuspended in buffer A (50 mM Tris–HCl pH 7.5, 200 mM NaCl, 5 mM imidazole) and lysed at 4°C using an Avestin EmulsiFlex-C3 homogenizer. The lysate was centrifuged at 48,000 × *g* for 45 min at 4°C, and the supernatant loaded onto a 5-ml HisTrap HP column (GE Healthcare) equilibrated in buffer A. The column was washed with 10 CV of buffer A before elution with 5%, 10%, 15%, 20%, and 50% of buffer B (50 mM Tris–HCl pH 7.5, 200 mM NaCl, 500 mM imidazole). TseV3:TsiV3 complexes were concentrated using a Vivaspin spin-concentrator and further purified by size-exclusion chromatography at 4°C on a Superdex 75 16/600 column (GE Healthcare) equilibrated in 25 mM Tris–HCl pH 7.5, 50 mM NaCl, and 5% glycerol. Eluted complexes were denatured in 6 M urea at 4°C for 16 hr, then loaded onto a 5 ml HisTrap column to remove the immunity protein (the same buffers were used but containing 6 M urea). The pooled protein fractions containing only 6xHis-TseV3 were concentrated to 0.2–0.7 g/l and diluted drop wise into 1 l of refolding buffer (50 mM Tris–HCl pH 7.5, 200 mM NaCl, 5% glycerol) using a peristaltic pump at 4°C under constant stirring. The refolded TseV3 was finally applied onto 5 ml HiTrap column to concentrate.

## Nuclease activity

For *in vitro* biochemical assays, oligonucleotides' sequences were retrieved from *Kratz et al., 2010*; *Figure 5—figure supplement 1*. PAGE purified DNA oligonucleotides were purchased from Thermo Fisher Scientific, in which F9 was labeled with FAM (6-Carboxyfluorescein) at the 5′ end. Oligonucleotides were annealed by mixing in 1:3 ratio F9-FAM and additional oligonucleotides (400:1200 nM) in buffer containing 25 mM Tris–HCl (pH 7.5), 50 mM NaCl, and heated for 5 min at 95°C prior to slow cooling to room temperature. Annealed substrates were checked in a native 20% polyacrylamide gel

separated in TBE (Tris/borate/EDTA) buffer, visualized in a Bio-Rad ChemiDoc Universal Hood III Gel Documentation System with Image Lab software.

For the endonuclease assay, 80 nM of labeled substrates were incubated with 800 nM of enzyme (TseV3$_{WT}$ or TseV3$_{D230A}$) in 25 mM Tris–HCl (pH 7.5), 50 mM NaCl, BSA 0.1 mg/ml and 5 mM MnCl$_2$ at final volume of 10 µl for 1 hr at 37°C. Reactions were stopped with 10 µl of stop buffer (50% forma-mide, 30 mM EDTA, 6% glycerol, 0.25% bromophenol blue), heated for 10 min at 95°C and sepa-rated on a 20% denaturing polyacrylamide gel with 7 M urea in TBE with warm buffer (50–60°C). Gels were visualized in a Bio-Rad ChemiDoc Universal Hood III Gel Documentation System with Image Lab software. To test the cofactors, reactions were carried as described before, but using only the splayed arm substrate and with the addition of 5 mM EDTA, 5 mM ZnCl$_2$ or 5 mM MgCl$_2$ as indicated. Time-course degradation was carried out with 160 nM of labeled splayed arm substrate preincubated with 1600 nM of TseV3$_{WT}$ for 10 min at 37°C in 25 mM Tris–HCl (pH 7.5), 50 mM NaCl, and BSA 0.1 mg/ml to allow binding to occur. The reaction was started by the addition of 5 mM MnCl$_2$ and stopped with stop buffer at the indicated time-points. Samples were heated for 10 min at 95°C and run on a denaturing 20% polyacrylamide gel with oligonucleotides of known sizes as markers (*Figure 5—figure supplement 1*) and visualized in a Bio-Rad ChemiDoc Universal Hood III Gel Documentation System with Image Lab software. The same gel was also silver stained to allow visualization of the markers. The ssDNA substrate used as control was a 59mer poly-T with a 5′-FAM to prevent the formation of secondary structures, which could be recognized by the enzyme.

For in vivo analysis of nuclease activity, *E. coli* DH5α harboring pBRA TseV2$_{WT}$, TseV2$_{D282A}$, and TseV3$_{WT}$ or TseV3$_{D230A}$ were subcultured in LB with 0.2% d-glucose (1:50) and grown at 37°C (180 rpm) until OD$_{600\ nm}$ 0.4–0.6. Cells were harvested and resuspended in new media with 0.2% l-arabinose and growth for an additional 1 or 2 hr. Plasmids were extracted from 4 ml of culture with OD$_{600\ nm}$ adjusted to 1 with GeneJET Plasmid Miniprep Kit (#K0503 Thermo Fisher Scientific) and separated on 1% agarose gel stained with Syber Safe using GeneRuler 1 kb (#SM0311 Thermo Fisher Scientific) as marker.

## Crystallography and structure determination

TseV3–TsiV3 was concentrated to 18 mg/ml and crystalized in 0.1 M HEPES (4-(2-hydroxyethyl)-1-pipe razineethanesulfonic acid) pH 7.5 and 30% (vol/vol) PEG Smear Low (12.5% [vol/vol] PEG 400, 12.5% [vol/vol] PEG 500, monomethylether, 12.5% [vol/vol] PEG 600, 12.5% [vol/vol] PEG 1000). The crystals were cryoprotected in the mother liquor supplemented with 20% ethylene glycol and subsequently cryo-cooled in liquid nitrogen. X-ray diffraction data were collected at Diamond Light Source on beamline i04, and initial data processing was performed using the xia2-dials pipeline (*Winter, 2010*; *Winter et al., 2018*). The data were phased by molecular replacement in Phaser (*McCoy et al., 2007*) (RRID:SCR_014219) using AlphaFold (*Jumper et al., 2021*) models of TseV3$_{134–289}$ and TsiV3$_{10–327}$, which were trimmed to include only the high-confidence regions and omit the N-terminal DUF4150 domain of TseV3.

## Quantification and statistical analyses

Statistical test, number of events, mean values, and standard deviations are reported in each figure legend accordingly. Statistical analyses were performed using GraphPad Prism5 software and signifi-cance is determined by the value of $p < 0.05$.

## Acknowledgements

We are grateful to Cristiane Rodrigues Guzzo for sharing reagents and equipment, and Alexandre Bruni Cardoso for allowing access to the fluorescence microscope. We thank members of the LEEP and EBSlab for scientific discussions. Crystallography analyses were performed at Diamond Light Source. This work was supported by São Paulo Research Foundation (FAPESP) grants to RFdS (2016/09047-8), CSF (2017/17303-7), and EBS (2017/02178-2). ALL is supported by Welcome Trust (209437/Z/17/Z). FAPESP fellowships were awarded to JTH (2018/25316-4), DESL (2019/22715-8), GGN (2021/03400-6), EEL (2019/12234-2), GSC (2020/15389-4), and EBS (2018/04553-8). LM is supported by a MIBTP studentship.

# Additional information

## Funding

| Funder | Grant reference number | Author |
| --- | --- | --- |
| Sao Paulo Research Foundation | 2016/09047-8 | Robson Francisco de Souza |
| Sao Paulo Research Foundation | 2017/17303-7 | Chuck Shaker Farah |
| Sao Paulo Research Foundation | 2017/02178-2 | Ethel Bayer-Santos |
| Wellcome Trust | 209437/Z/17/Z | Andrew L Lovering |
| FAPESP Fellowship | 2018/25316-4 | Julia Takuno Hespanhol |
| FAPESP Fellowship | 2019/22715-8 | Daniel Enrique Sanchez-Limache |
| FAPESP Fellowship | 2021/03400-6 | Gianlucca Gonçalves Nicastro |
| FAPESP Fellowship | 2019/12234-2 | Edgar Enrique Llontop |
| FAPESP Fellowship | 2020/15389-4 | Gustavo Chagas-Santos |
| FAPESP Fellowship | 2018/04553-8 | Ethel Bayer-Santos |
| MIBTP Studentship | | Liam Mead |

The funders had no role in study design, data collection, and interpretation, or the decision to submit the work for publication. For the purpose of Open Access, the authors have applied a CC BY public copyright license to any Author Accepted Manuscript version arising from this submission.

## Author contributions

Julia Takuno Hespanhol, Conceptualization, Data curation, Formal analysis, Validation, Investigation, Visualization, Methodology, Writing – original draft, Writing – review and editing; Daniel Enrique Sanchez-Limache, Conceptualization, Formal analysis, Validation, Investigation, Visualization, Methodology; Gianlucca Gonçalves Nicastro, Data curation, Software, Formal analysis, Investigation, Visualization, Methodology; Liam Mead, Data curation, Formal analysis, Validation, Investigation, Visualization, Methodology; Edgar Enrique Llontop, Formal analysis, Investigation, Visualization, Methodology; Gustavo Chagas-Santos, Formal analysis, Validation, Investigation, Methodology; Chuck Shaker Farah, Rodrigo da Silva Galhardo, Resources, Supervision, Funding acquisition, Methodology; Robson Francisco de Souza, Resources, Software, Supervision, Funding acquisition, Methodology; Andrew L Lovering, Resources, Data curation, Formal analysis, Supervision, Funding acquisition, Validation, Investigation, Visualization, Methodology; Ethel Bayer-Santos, Conceptualization, Resources, Data curation, Formal analysis, Supervision, Funding acquisition, Validation, Investigation, Visualization, Methodology, Writing – original draft, Project administration, Writing – review and editing

## Author ORCIDs

Julia Takuno Hespanhol ![ORCID] http://orcid.org/0000-0001-6701-7286
Daniel Enrique Sanchez-Limache ![ORCID] http://orcid.org/0000-0003-1364-7527
Edgar Enrique Llontop ![ORCID] http://orcid.org/0000-0003-4910-9667
Robson Francisco de Souza ![ORCID] http://orcid.org/0000-0002-5284-4630
Rodrigo da Silva Galhardo ![ORCID] http://orcid.org/0000-0002-5686-9704
Ethel Bayer-Santos ![ORCID] http://orcid.org/0000-0003-3832-3449

## Decision letter and Author response

Decision letter https://doi.org/10.7554/eLife.82437.sa1
Author response https://doi.org/10.7554/eLife.82437.sa2

# Additional files

## Supplementary files

• Supplementary file 1. Amino acid sequence alignment of VgrGs. (A) Amino acid sequence alignment of VgrG1, VgrG2, and VgrG3. (B) Amino acid sequence alignment of VgrG2 and VgrG3. Amino acids are color-coded according to their properties.

• Supplementary file 2. List of all homologs collected by JackHMMER searches and used to build the phylogenetic tree shown in *Figure 3A*.

• Supplementary file 3. Genomic context of members of each VRR-Nuc subfamily.

• Supplementary file 4. Plasmidial DNA extraction from *E. coli* expressing TseV2, TseV2$_{D282A}$, TseV3, or TseV3$_{D230A}$ for 1–2 hr visualized on 1% agarose gel.

• Supplementary file 5. Amino acid sequence alignment of TseV1-4 and P. aeruginosa TseV (PA0822) based on secondary structures.
 (A) Manual amino acid sequence alignment of TseV14 and *P. aeruginosa* TseV (PA0822) based on secondary structures. The secondary structures are indicated above the alignments with α-helixes represented by spirals and β-sheets by arrows. The conserved catalytic residues are highlighted in red with the logo underneath the alignments. TseV4 contains another start codon located upstream of the annotated one. (B) TseV1–4 structures predicted by the AlphaFold (*Jumper et al., 2021*). Underneath is the conserved PD-(D/E)xK enzymatic core with the absent structures marked in dashed red.

• Supplementary file 6. Crystallographic statistics of the TseV3:TsiV3 complex.

• Supplementary file 7. List of strains, plasmids and primers used in the study.

• MDAR checklist

## Data availability

All data generated during this study are included in the manuscript and supporting files. Source data files have been provided.

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
