## [Editor Report]

This paper will be of interest to microbiologists studying the molecular mechanisms by which bacteria compete with one another, bacterial physiology and toxins in general. Hespanhol et al. report here the characterization of the antibacterial VRR-Nuc family of type VI secretion system effectors that are endonucleases that display antibacterial activity by inducing DNA double-strand breaks.

---

## [Decision Letter]

**Decision letter after peer review:**

[Editors’ note: the authors resubmitted a revised version of the paper for consideration. What follows is the authors’ response to the first round of review.]

Thank you for submitting the paper "Antibacterial T6SS effectors with a VRR-Nuc domain induce target cell death via DNA Double-Strand Breaks" for consideration by *eLife*. Your article has been reviewed the evaluation has been overseen by a Reviewing Editor and a Senior Editor.

Comments to the Authors:

We are sorry to say that, after consultation, we have decided that this work will not be considered further for publication by *eLife*.

Specifically, this manuscript requires protein biochemistry to be considered a complete and novel study. This effector family has the potential to be interesting but needs evidence from biochemical assays using purified protein to determine what this protein family actually does. If this effector family has a unique biochemistry that is distinct from previously published T6SS DNase effectors, this manuscript would make an excellent contribution.

The suggested essential revisions are extensive and beyond what is normally considered reasonable. Some of the suggested experiments may already be in progress by the authors. For these reasons, we reject this full submission but leave open the possibility of a re-submission of a novel manuscript that would be considered as revision and for which we would make an effort to be reviewed by the same individuals.

Essential revisions:

1) The lack of protein-nucleic acid biochemistry in this manuscript makes some of the stated conclusions untrue. For example, the title of the manuscript states that these effectors cause DNA double-stranded breaks but there no direct evidence of this. The DAPI staining and SOS reporter data are interesting preliminary data, but protein biochemistry that directly demonstrates the proposed activity by the toxin is lacking. The authors should use purified wild type and catalytic mutant proteins to show DSB formation *in vitro*.

2) The GamGFP data are interesting but could reflect a downstream consequence of the toxin's activity rather than being directly due the toxin itself. The authors note that they do not see any obvious degradation of genomic DNA (though this data are not shown and probably should be shown). Even if the bacterial chromosome is only hydrolyzed at a small number of sites as the authors propose, this should be obvious by gel electrophoresis. DSB formation in cells should be directly demonstrated.

3) The structural biology section speculates about inhibition mechanism and DNA binding. Biochemical analysis of the TsiV3-L131K and TseV3-P256K showing defective complex formation would significantly strengthen the validation of the structural model.

4) In Figure 1C, the wild-type competition experiment does not appear to have any error associated with it. This seems unusual and the authors may want to compare their methodology with what is typically done in the T6SS field. Additionally, the magnitudes of the reported competition phenotypes throughout the manuscript (ranging from 1.5-2.0 fold) are quite small compared to almost all other characterized T6SSs.

Orthogonal methods to show that TseV2 and TseV3 are secreted from S. bongori in a T6SS-dependent manner would strengthen the conclusion.

Prey recovery is normalized to wildtype. Please add information in the legend or text, what the recovery rate is in wildtype, in other words how effective is the anti-bacterial activity? This applies also to Figure 2E, F.

5) In Figure 2E and Figure 2F, the differences are small (~2 fold or less) and the authors use a small number of biological replicates (N = 3 – 4). Although the differences are statistically significant, one concern is the reproducibility of these small phenotypes, especially if there is a possibility of small differences in the conditions used during experimentation. The author's conclusions from these data would be strengthened if the authors repeated these experiments to demonstrate the reproducibility of these findings. This is important given that this is the first time these T6SS effectors have been described to have antibacterial activity.

6) Figure 3D lacks the control of no TseV2 or 3 to compare the catalytic mutant with the absence of the effector protein.

7) Figure 4: Add controls to express Ts3V2 and 3 catalytic mutants in cells with Gam-GFP and recA::GFP.

In Figures 4A and Figure 4B, GFP fluorescence signal can be influenced by other factors (such as oxygen). Using alternative methods such as RNA or protein levels to measure expression of recA would strengthen the author's conclusions.

---

## [Author Response]

[Editors’ note: the authors resubmitted a revised version of the paper for consideration. What follows is the authors’ response to the first round of review.]

Essential revisions:1) The lack of protein-nucleic acid biochemistry in this manuscript makes some of the stated conclusions untrue. For example, the title of the manuscript states that these effectors cause DNA double-stranded breaks but there no direct evidence of this. The DAPI staining and SOS reporter data are interesting preliminary data, but protein biochemistry that directly demonstrates the proposed activity by the toxin is lacking. The authors should use purified wild type and catalytic mutant proteins to show DSB formation *in vitro*.

We have included the required protein biochemistry in the revised manuscript (Figure 5). In this study, we show that TseVs effectors are homologs of the human FAN1 and *P. aeruginosa Pa*FAN1, which also contain the VRR-Nuc domain. FAN1 and *Pa*FAN1 function in interstrand DNA crosslink repair pathways and were shown to be structure-specific endonucleases that cleave 5’ flap substrates (Gwon *et al.*, 2014; Kratz *et al.*, 2010; MacKay *et al.*, 2010; Pennell *et al.*, 2014; Smogorzewska *et al.*, 2010). Here we show that the TseV3 effector is also a structure-specific endonuclease that preferentially cleave splayed arms substrates (Figure 5). Previous work showed that FAN1 and *Pa*FAN1 recognize the Y shape 5’ flap substrate cleaving a couple of nucleotides (1-5 nt) downstream from the arms junction, and later cleaving the opposite strand to create a double-strand end (Gwon *et al.*, 2014; Kratz *et al.*, 2010; MacKay *et al.*, 2010; Pennell *et al.*, 2014; Smogorzewska *et al.*, 2010). According to the size of the fragments generated after cleavage of the 60-mer oligonucleotide F9 by TseV3 (Figure 5), the effector displays endonuclease activity and cleaves the slayed arm substrate in the dsDNA region. Given the DNA structure that is degraded by TseV3, we hypothesized that it acts on the replication forks or transcription bubbles of target cells. Hence, the effector activity could be directly responsible for creating the double-strand breaks detected in the GamGFP reporter assay, which can detect up to a four-base single-strand DNA overhang (Akroyd and Symonds, 1986; Shee *et al.,* 2013). In addition, TseV3 activity on replication forks could promote the collapse of the replication machinery, inducing double-strand breaks (Kuzminov, 1999). This information was discussed in the new version of the manuscript.

2) The GamGFP data are interesting but could reflect a downstream consequence of the toxin's activity rather than being directly due the toxin itself. The authors note that they do not see any obvious degradation of genomic DNA (though this data are not shown and probably should be shown). Even if the bacterial chromosome is only hydrolyzed at a small number of sites as the authors propose, this should be obvious by gel electrophoresis. DSB formation in cells should be directly demonstrated.

We included images of agarose gels showing the small degradation observed in plasmids after extraction from *E. coli* expressing the wild-type and catalytic mutant (Supplementary File 4 – revised manuscript). As described above (#1), we believe that the biochemistry assays with purified recombinant protein indicating that TseV3 cleaves splayed arms in the double-strand region (Figure 5 – revised version) provides support to the results obtained with the GamGFP assay (Figure 4G-H). However, recognizing the possibility that double-strand breaks could also be created by the collapse of the replication fork, we decided to tone down the initial claim, and discuss about these possibilities in the revised manuscript.

3) The structural biology section speculates about inhibition mechanism and DNA binding. Biochemical analysis of the TsiV3-L131K and TseV3-P256K showing defective complex formation would significantly strengthen the validation of the structural model.

We expressed individual proteins and performed affinity chromatography and size-exclusion chromatography, which revealed the formation of an effector-immunity complex. Although unexpected, these results show that mutated proteins were still able to interact, but the interference caused by point mutations was sufficient to prevent neutralization of the effector in toxicity assays. These data were included in Figure 6 supplement 1B and discussed in the revised version of the manuscript.

4) In Figure 1C, the wild-type competition experiment does not appear to have any error associated with it. This seems unusual and the authors may want to compare their methodology with what is typically done in the T6SS field. Additionally, the magnitudes of the reported competition phenotypes throughout the manuscript (ranging from 1.5-2.0 fold) are quite small compared to almost all other characterized T6SSs.

We are aware of the methodology used for other bacterial species in the T6SS field, but we decided to use the “prey recovery rate” readout for *Salmonella* species as its T6SSs do not appear to be fully activated in the *in vitro* conditions used for interbacterial competition experiments. This is a topic that is currently under investigation in the lab. In the meantime, we are working with basal levels of expression, and we have standardized to represent the output of bacterial competition assays as “prey recovery rate”, which is normalized to wild-type values that are equal 1. This is the reason why there is no error bars in wild-type samples. A similar approach was used in a previous work from our group (Sibinelli-Sousa *et al.,* 2020). Although the differences between wild-type and mutants are small, these are reproducible. We have included another 3 replicates in Figure 1C of the revised manuscript, thus comprising a total of six independent experiments with similar results. It is worth highlighting that the small differences in prey recovery rate detected between wild-type and Δ*tssB* are readily restored after complementation of the mutant with a plasmid. The raw data containing CFU numbers are available in the source data file.

Orthogonal methods to show that TseV2 and TseV3 are secreted from S. bongori in a T6SS-dependent manner would strengthen the conclusion.

The low expression of T6SS components in the conditions used for interbacterial competition does not allow us to detect reasonable amounts of secreted proteins in the supernatant. We believe that the interbacterial competition experiments shown in Figure 2E provide convincing evidence that TseV2 and TseV3 are secreted via the SPI-22 T6SS. We have included another 3 replicates in Figure 2E of the revised manuscript, comprising a total of six independent experiments with similar results. It is worth highlighting that the small differences in prey recovery rate detected between wildtype and Δ*tssB* are readily restored after complementation of the mutant with a plasmid encoding the cognate immunity protein. The raw data containing CFU numbers are available in the source data file.

Prey recovery is normalized to wildtype. Please add information in the legend or text, what the recovery rate is in wildtype, in other words how effective is the anti-bacterial activity? This applies also to Figure 2E, F.

The raw data containing CFU numbers for Figure 1C, Figure 2E and 2F are available in the source data file submitted with the manuscript. As mentioned in response to #4.1, the T6SSs of *Salmonella* spp. do not seem to be fully activated in the *in vitro* conditions used for interbacterial competition experiments, and this topic is currently under investigation in the lab. Our hypothesis is that the *Salmonella* spp. T6SSs are mainly defensive systems in oppose to other more “potent” systems which are offensive systems that are constitutively active. Nevertheless, the basal levels of expression of SPI22 T6SS of *S. bongori* allowed us to determine its antibacterial activity. Besides the differences detected between WT and Δ*tssB* in interbacterial competitions, we were able to detect basal levels of SPI-22 T6SS expression in LB media by tagging the Hcp with a C-terminal FLAG-tag in the bacterial chromosome, which is regulated by endogenous promoters (data not shown). Although small levels of Hcp-FLAG were detected in total protein extracts, this was not enough to be detect in the supernatant (#4.2). As we are working in suboptimal conditions of SPI-22 T6SS activation, we do not see prudent to compare and discuss the effectiveness of its activity with other bacterial species.

5) In Figure 2E and Figure 2F, the differences are small (~2 fold or less) and the authors use a small number of biological replicates (N = 3 – 4). Although the differences are statistically significant, one concern is the reproducibility of these small phenotypes, especially if there is a possibility of small differences in the conditions used during experimentation. The author's conclusions from these data would be strengthened if the authors repeated these experiments to demonstrate the reproducibility of these findings. This is important given that this is the first time these T6SS effectors have been described to have antibacterial activity.

We have included another 3 replicates in Figure 2E and 2F of the revised manuscript, comprising a total of six independent experiments with similar results. The raw data containing CFU numbers are available in the source data file. It is worth mentioning that despite the small fold-changes shown in Figure 2E and Figure 2F, results show that TseV2 and TseV3 require different VgrG proteins for secretion – VgrG2 and VgrG3, respectively. This difference in VgrG requirement strengthens the reproducibility and specificity of the results.

6) Figure 3D lacks the control of no TseV2 or 3 to compare the catalytic mutant with the absence of the effector protein.

We included the required control in Figure 3D.

7) Figure 4: Add controls to express Ts3V2 and 3 catalytic mutants in cells with Gam-GFP and recA::GFP.In Figures 4A and Figure 4B, GFP fluorescence signal can be influenced by other factors (such as oxygen). Using alternative methods such as RNA or protein levels to measure expression of recA would strengthen the author's conclusions.

The experiments were repeated to include the catalytic mutants TseV2_D282A_ and TseV3_D230A_ (Figure 4A, 4B, 4G and 4H – revised manuscript). Western blots with anti-GFP and anti-DNAk antibodies were used to confirm GFP expression levels (Figure 4 —figure supplement 1).

References

Akroyd J, Symonds N (1986) Localization of the gam gene of bacteriophage mu and characterisation of the gene product. *Gene* 49: 273-282

Gwon GH, Kim Y, Liu Y, Watson AT, Jo A, Etheridge TJ, Yuan F, Zhang Y, Kim Y, Carr AM (2014) Crystal structure of a Fanconi anemia-associated nuclease homolog bound to 5′ flap DNA: basis of interstrand cross-link repair by FAN1. *Genes and development* 28: 2276-2290

Jin H, Roy U, Lee G, Schärer OD, Cho Y (2018) Structural mechanism of DNA interstrand cross-link unhooking by the bacterial FAN1 nuclease. *Journal of Biological Chemistry* 293: 6482-6496

Kosinski J, Feder M, Bujnicki JM (2005) The PD-(D/E) XK superfamily revisited: identification of new members among proteins involved in DNA metabolism and functional predictions for domains of (hitherto) unknown function. *BMC bioinformatics* 6: 1-13

Kratz K, Schöpf B, Kaden S, Sendoel A, Eberhard R, Lademann C, Cannavó E, Sartori AA, Hengartner MO, Jiricny J (2010) Deficiency of FANCD2-associated nuclease KIAA1018/FAN1 sensitizes cells to interstrand crosslinking agents. *Cell* 142: 7788

Kuzminov A (1999) Recombinational repair of DNA damage in *Escherichia coli* and bacteriophage λ. *Microbiology and molecular biology reviews* 63: 751-813

Liu T, Ghosal G, Yuan J, Chen J, Huang J (2010) FAN1 acts with FANCI-FANCD2 to promote DNA interstrand cross-link repair. *Science* 329: 693-696

MacKay C, Déclais A-C, Lundin C, Agostinho A, Deans AJ, MacArtney TJ, Hofmann K, Gartner A, West SC, Helleday T (2010) Identification of KIAA1018/FAN1, a DNA repair nuclease recruited to DNA damage by monoubiquitinated FANCD2. *Cell* 142: 65-76

Pennell S, Déclais A-C, Li J, Haire LF, Berg W, Saldanha JW, Taylor IA, Rouse J, Lilley DM, Smerdon SJ (2014) FAN1 activity on asymmetric repair intermediates is mediated by an atypical monomeric virus-type replication-repair nuclease domain. *Cell reports* 8: 84-93

Pizzolato J, Mukherjee S, Schärer OD, Jiricny J (2015) FANCD2-associated nuclease 1, but not exonuclease 1 or flap endonuclease 1, is able to unhook DNA interstrand cross-links in vitro. *Journal of Biological Chemistry* 290: 22602-22611

Shee C, Cox BD, Gu F, Luengas EM, Joshi MC, Chiu L-Y, Magnan D, Halliday JA, Frisch RL, Gibson JL (2013) Engineered proteins detect spontaneous DNA breakage in human and bacterial cells. *Elife* 2: e01222

Sibinelli-Sousa S, Hespanhol JT, Nicastro GG, Matsuyama BY, Mesnage S, Patel A, de Souza RF, Guzzo CR, Bayer-Santos E (2020) A family of T6SS antibacterial effectors related to l, d-transpeptidases targets the peptidoglycan. *Cell reports* 31: 107813

Smogorzewska A, Desetty R, Saito TT, Schlabach M, Lach FP, Sowa ME, Clark AB, Kunkel TA, Harper JW, Colaiácovo MP (2010) A genetic screen identifies FAN1, a Fanconi anemia-associated nuclease necessary for DNA interstrand crosslink repair. *Molecular cell* 39: 36-47

Wang R, Persky NS, Yoo B, Ouerfelli O, Smogorzewska A, Elledge SJ, Pavletich NP (2014) Mechanism of DNA interstrand cross-link processing by repair nuclease FAN1. *Science* 346: 1127-1130

Zhao Q, Xue X, Longerich S, Sung P, Xiong Y (2014) Structural insights into 5′ flap

DNA unwinding and incision by the human FAN1 dimer. *Nature communications* 5: 1-9